# The skeletal muscle response to high-intensity training assessed by single-nucleus RNA-sequencing is blunted in individuals with type 2 diabetes

Maria Hansen[1] 🆔, Julius E. R. Grothen[2,3] 🆔, Anders Karlsen[1], Jaime M. Martinez[4] 🆔, Nikos Sidiropoulos[4], Jørn W. Helge[5], Thomas Å. Pedersen[2] and Flemming Dela[1,6] 🆔

[1]*Xlab, Department of Biomedical Sciences, Faculty of Health and Medical Sciences, University of Copenhagen, Copenhagen, Denmark*

[2]*Global Drug Discovery, Novo Nordisk A/S, Copenhagen, Denmark*

[3]*Novo Nordisk Foundation Center for Basic Metabolic Research, Faculty of Health and Medical Sciences, University of Copenhagen, Copenhagen, Denmark*

[4]*Computational Biology, AI & Digital Research, Novo Nordisk A/S, Copenhagen, Denmark*

[5]*Department of Biomedical Sciences, Faculty of Health and Medical Sciences, University of Copenhagen, Copenhagen, Denmark*

[6]*Laboratory of Sports and Nutrition Research, Riga Stradins University, Riga, Latvia*

Handling Editors: Paul Greenhaff & Bettina Mittendorfer

The peer review history is available in the Supporting Information section of this article (https://doi.org/10.1113/JP288368#support-information-section).

**Abstract figure legend** The figure illustrates the impact of high-intensity interval training (HIIT) on skeletal muscle metabolism and insulin sensitivity in individuals with type 2 diabetes and healthy controls. HIIT consists of short bursts of intense exercise followed by rest periods. The study design involved one-legged HIIT, skeletal muscle biopsies, single-nucleus RNA sequencing (snRNA-seq) and immunofluorescence to analyse metabolic changes. Key findings show that HIIT improved insulin sensitivity in both groups, but individuals with type 2 diabetes exhibited a blunted myonuclear metabolic transcriptional response. In healthy controls, HIIT increased glycolysis and glycogen breakdown, whereas these responses were diminished in type 2 diabetes. Additionally, immunofluorescence significantly correlated with the proportion of myonuclei in the snRNA-seq analysis. These findings highlight differences in skeletal muscle responses to HIIT between individuals with and without type 2 diabetes, providing insights into disease-related impairments in exercise adaptation.

The Journal of Physiology

**Abstract**  Training can improve insulin sensitivity in individuals with type 2 diabetes, but a clear understanding of the mechanisms remains elusive. To further our knowledge in this area, we aimed to examine the effect of type 2 diabetes and of high-intensity interval training (HIIT) on the nuclear transcriptional response in skeletal muscle. We performed single-nucleus RNA-sequencing (snRNA-seq) and immunofluorescence analysis on muscle biopsies from the trained and the untrained legs of participants with and without type 2 diabetes, after 2 weeks of one-legged HIIT on a cycle ergometer. Surprisingly, the type 2 diabetes condition only seemed to have a minor effect on transcriptional activity in myonuclei related to major metabolic pathways when comparing the untrained legs. However, while in particular the type IIA myonuclei in the control group displayed a considerable metabolic response to HIIT, with increases in genes related to glycogen breakdown and glycolysis primarily in the type IIA myonuclei of the trained leg, this response was blunted in the diabetes group, despite a marked increase in glucose clearance in both groups. Additionally, we observed that fibre type distribution assessed by immunofluorescence significantly correlated with the proportion of myonuclei in the snRNA-seq analysis. In conclusion, the type 2 diabetes condition blunts the metabolic transcriptional response to HIIT in the type IIA myonuclei without affecting the improvement in insulin sensitivity. Additionally, our results indicate that snRNA-seq can be used as a surrogate marker for fibre type distribution in sedentary middle-aged adults.

(Received 17 December 2024; accepted after revision 17 April 2025; first published online 14 May 2025)

**Corresponding author** Flemming Dela: Xlab, Department of Biomedical Sciences, University of Copenhagen, Blegdamsvej 3, 2200 Copenhagen N, Denmark.    Email: fdela@sund.ku.dk

**Key points**

- The study utilized single-nucleus RNA sequencing (snRNA-seq) to analyse 38 skeletal muscle biopsies, revealing distinct transcriptional profiles in myonuclei from individuals with and without type 2 diabetes (T2D) after 2 weeks of HIIT.
- snRNA-seq identified significant differences in gene expression, with 14 differentially expressed genes (DEGs) in type IIA myonuclei of the control group, specifically related to glycogen breakdown and glycolysis, which were blunted in the T2D group.
- In the control group, HIIT induced a substantial transcriptional response in type IIA myonuclei, enhancing metabolic pathways associated with insulin sensitivity, while the T2D group showed minimal transcriptional changes despite improved insulin sensitivity.
- The T2D group exhibited a blunted response in metabolic gene expression, indicating that the training effect on muscle adaptation was significantly impaired compared to healthy controls.
- Overall, the findings highlight the differential impact of HIIT on muscle metabolism, emphasizing the need for tailored exercise interventions for individuals with T2D.

# Introduction

Type 2 diabetes is estimated to affect half a billion people around the world (International Diabetes Federation, 2021) and is associated with an increased risk of cardiovascular complications such as ischaemic heart disease and stroke (Emerging Risk Factors Collaboration, 2010). Impaired insulin sensitivity (i.e. insulin resistance) is

**Maria Hansen** completed medical school in 2018 from the University of Copenhagen, where she is currently concluding her PhD at the Department of Biomedical Sciences under the supervision of Professor Flemming Dela. Her research focuses on type 2 diabetes and aims to elucidate the mechanisms behind the improvements in insulin sensitivity observed after training. To this end, she has investigated insulin sensitivity, metabolic flexibility, mitochondrial respiration and myonuclear transcriptional changes in participants with and without type 2 diabetes before and after training.

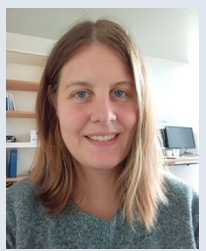

a major characteristic of type 2 diabetes and plays a significant role in the hyperglycaemia seen in these individuals (DeFronzo et al., 2015).

Training can improve insulin sensitivity in individuals with type 2 diabetes (Dela et al., 1995; Holten et al., 2004; Krotkiewski et al., 1985), but some patients who engage in exercise may fail to achieve the anticipated improvements in glycaemic control (Bouchard et al., 2012; Stephens & Sparks, 2015). Increasing the exercise intensity (Hrubeniuk et al., 2024) does not seem to alleviate the problem. However, while most individuals with type 2 diabetes experience training-induced improvements in insulin sensitivity and glycaemic control, the underlying mechanisms remain elusive, though cellular adaptations like translocation of the glucose transporter GLUT4 (Christ-Roberts et al., 2004; Dela et al., 1994; Hussey et al., 2011) have been implicated. With the emergence of more advanced methods to examine the cellular machinery, progress may be on the way.

With single-nucleus RNA-sequencing (snRNA-seq), there is an opportunity to bypass some of the challenges of other types of RNA-sequencing since snRNA-seq can examine transcriptional differences related to specific nuclei and is not restricted by cell size or multinucleation (Williams et al., 2022). The snRNA-seq method is gaining in frequency and has recently been used on human skeletal muscle in published studies in the contexts of healthy individuals (Karlsen et al., 2023; Nieves-Rodriguez et al., 2023), ageing (Lai et al., 2024; Perez et al., 2022), peripheral artery disease (Pass et al., 2023) and myopathies (Casal-Dominguez et al., 2023; Nelke et al., 2023; Pinal-Fernandez et al., 2023; Soule et al., 2023; Suárez-Calvet et al., 2023; Xie et al., 2024), but not type 2 diabetes and not in response to training.

In the present study, we examined the transcriptional differences in single myonuclei between individuals with and without type 2 diabetes in an untrained state, and the response to a high-intensity interval training (HIIT) intervention in each group. To this end, we obtained skeletal muscle biopsies from the trained and untrained leg of all participants after 2 weeks of one-legged HIIT (Dela et al., 2019) and performed snRNA-seq on 38 biopsies. Additionally, we provide immunostainings to confirm the fibre type distribution of the sequenced myonuclei. These data provide a unique atlas of the transcriptional landscape in insulin resistant skeletal muscle and map the changes in response to a well-known intervention for improving insulin-mediated glucose clearance.

## Methods

### Study design

Muscle biopsies were collected and used in a previous study (Dela et al., 2019). In summary, ten sedentary males with type 2 diabetes and ten healthy males matched for age, body mass index (BMI) and peak oxygen uptake ($\dot{V}_{O_2peak}$) performed 2 weeks (eight sessions) of one-legged HIIT on a bicycle ergometer, with all participants being randomly allocated into training with the right or left leg. Each session started with a 2-min warm-up and included ten 1-min intervals, interspersed with 1 min of rest. The intervals started at 70% of the maximal one-legged workload and increased until obtaining a heart rate >80% of the maximal heart rate during the last two intervals. Forty hours after the last exercise bout, lateral vastus muscle biopsies from the trained and untrained legs were obtained, and a two-step isoglycaemic hyperinsulinaemic clamp (insulin infusion rates of 80 and 400 mU/min/m², 2 h per step) was performed. Participants were instructed to pause medication intake on the test day (1 week before for statin treatment) and to ingest a diet with a minimum of 250 g of carbohydrates in the preceding 3 days. Leg glucose clearance was calculated as plasma flow (blood flow × (1 − arterial haematocrit)) × ((arterial − venous)/arterial) glucose concentrations divided by leg muscle mass (kg).

The study was approved by the Danish National Committee on Health Research Ethics (H-4-2011-137) and conformed with the rules of the Helsinki Declaration, except for registration in a database. All participants signed a declaration of informed consent before inclusion.

### Muscle biopsies

The muscle biopsies were obtained using the Bergström technique, and visible connective tissue was immediately removed. One portion of the muscle was snap-frozen in liquid nitrogen, while another was embedded in OCT/Tissue-Tek and snap-frozen in isopentane precooled in liquid nitrogen. The samples were stored at −80°C until analysis.

### Nuclei isolation and counting

Nuclei from 38 available biopsies (from ten participants without and nine participants with type 2 diabetes) were isolated using a modified version of the Nuclei Isolation Kit: Nuclei PURE Prep, cat. no. NUC201 from Sigma–Aldrich. The entire protocol was carried out on ice, and the protocol used, on average, 68 mg of muscle tissue (range 20–115 mg), which yielded an average of $1.9 \times 10^5$ nuclei per sample (range $7.5 \times 10^4$–$2.5 \times 10^5$ nuclei per sample). The tissue was lysed in 2 ml of ice-cold lysis buffer (Nuclei Pure Lysis Buffer, Sigma cat. no. L9286, with 1 mM DL-dithiothreitol (DTT), Sigma cat. no. D9779, and 0.01% Triton X-100, Sigma cat. no. T1565), briefly chopped into pieces with a scalpel, and homo-

genized for 30 s using an ULTRA-TURRAK TP18/10 homogenizer (20,000 rpm). The homogenized tissue was then lysed for 10 min before passing the lysate through a 40 μm cell strainer and collecting it in Eppendorf DNA LoBind tubes. The nuclei were then pelleted by centrifugation at 500 × *g* for 10 min at 4°C. Following centrifugation, the supernatant was carefully removed, and the pellet was resuspended in wash buffer (1× DPBS, no calcium, no magnesium, Gibco cat. no. 14190, with 1% Human Serum Albumin Sigma cat. No. A1887, and 0.2 u/μl RNasin Plus RNase Inhibitor [40 u/μl], Promega cat. No. 2611). The process of filtering, centrifugation and resuspending was repeated once, followed by a wash without filtering for a total of three washes. The washed filtered lysate was then mixed with 1.8 M Sucrose Cushion Solution (mix of Nuclei PURE 2 M Sucrose Cushion Solution, Sigma cat. no. S9308, Nuclei PURE Sucrose Cushion Solution, Sigma cat. no. S9058 and 0.83 mM DTT) for a sucrose solution of approximately 1.16 M sucrose. The mixture was then carefully loaded on top of 500 μl 1.8 M Sucrose Cushion Solution, and centrifuged at 13,000 × *g*, 4°C for 45 min. The supernatant was carefully removed, and the nuclei-containing pellet was resuspended in wash buffer, centrifuged at 500 × *g* for 10 min at 4°C, resuspended in wash buffer and passed through a 20 μm filter. The resuspension in wash buffer, filtering and centrifugation was then repeated once for a total of two washes. Finally, the pellet was resuspended in approximately 200 μl wash buffer and counted using NucleoCounter NC-200 (ChemoMetec cat. no. 900-0201) to determine nuclei concentration.

## Single-nucleus RNA-seq using Chromium Next GEM Single Cell

For isolation of single-nucleus RNA, the samples were run using the 10X Genomics Chromium Next GEM Single Cell 3′ Reagent Kits v3.1 on a 10X Chromium Controller according to manufacturer's instructions (protocol version 'CG000204_ChromiumNextGEMSingleCell3_v3.1_Rev_D'). Before loading the samples, the nuclei were diluted to below 1200 nuclei/μl. The average concentration of nuclei was ∼950 nuclei/μl (range 375–12,400 nuclei/μl) and ∼8300 nuclei were loaded from each sample for an expected recovery of ∼5000 nuclei per sample. During the run, the sample cDNA quality control and quantification were assessed using a Bioanalyzer High Sensitivity DNA kit (Agilent cat. no. 5067-4626). Post run, the Agilent Bioanalyzer High Sensitivity DNA kit was used to estimate fragment size coupled with a concentration measurement on a Qubit 4 Fluorometer (Invitrogen cat. no. Q33238) using the Qubit 1X dsDNA HS Assay Kit (Invitrogen cat. no. Q33230).

## NextSeq 550 sequencing

The libraries were sequenced on an Illumina NextSeq550 using NextSeq 500/550 High Output Kit v2.5 (150 Cycles) with up to 400M reads according to the manufacturer's instructions. The nuclei were loaded with one NextSeq550 kit per participant, thus loading the untrained and trained sample on the same chip. There were 200M reads per sample. The sequencing was done with 28 cycles (10X Index and UMI), 8 cycles (i7 Index), 0 cycles (i5 Index) and 91 cycles for the reads.

## Single-nucleus RNA-seq analysis

Raw BCL files were demultiplexed with the cellranger mkfastq software (v4.0.0) (Zheng et al., 2017). The demultiplexed libraries were then aligned to the human assembly GRCh38.p13 genome (National Library of Medicine, 2019) and counted using Cell ranger with pre-mRNA intronic regions included in the library (10X Genomics 2020).

The mapped cellranger count matrices were decontaminated for ambient RNA using SoupX (Young & Behjati, 2020) with the automatically detected threshold. For quality control, we removed cells which identified as outliers in either a high percentage of mitochondrial gene counts, a low number of features or low total counts. The quality control was run in R (v 4.1.0) using scran 1.22.1 (Lun et al., 2016), scater_1.22.0 and scuttle_1.4.0 (McCarthy et al., 2017). The adjusted count matrices were then used as the basis for the forward analysis in R (v 4.2.0) using Seurat (v4) (Hao et al., 2021). During the initial clustering of the data, it was clear that two clusters contained the majority of doublets due to the presence of multiple marker genes. These two clusters were removed manually prior to down-stream analysis.

For each individual sample library, we ran normalization (NormalizeData), feature selection (FindVariableFeatures), scaling (ScaleData) and PCA (RunPCA) separately before hierarchically merging the datasets using Seurat. The integration was done using the Seurat function FindIntegrationAnchors and IntegrateData. First, the data were integrated into groups based on training status and whether the samples were from the control or diabetes group. Then, the control samples and diabetes samples were merged separately. Finally, all the data were integrated for joint downstream analysis.

Using the Seurat functions FindNeighbors and FindClusters, the combined data set was clustered, and using RunUMAP and DimPlot, the dataset was visualized as a UMAP. Cluster labels were assigned using the Seurat function FindAllMarkers and then manually curated.

**Table 1. Antibodies for immunofluorescence**

| Antibodies for immunofluorescence | Dilution |
|---|---|
| **Primary antibodies** | |
| Laminin (mouse anti-laminin gamma 1; gene name *LAMC1*; IgG2a; #2E8; DSHB[b]) | 1:15 |
| MyHC-I (mouse anti-myosin heavy chain Type I; gene name *MYH7*; IgG2b; #2E8; DSHB[a]) | 1:15 |
| MyHC-IIA (mouse anti-myosin heavy chain Type IIA*; gene name *MYH2*, IgG1; #SC-71; DSHB[a]) | 1:15 |
| **Secondary antibodies** | |
| Alexa Fluor 488 goat anti-mouse IgG1; A-21121, ThermoFischer | 1:500 |
| Alexa Fluor 568 goat anti-mouse IgG2b; A-21144, ThermoFischer | 1:500 |
| Alexa Fluor 647 goat anti-mouse IgG2a; A-21241, ThermoFischer | 1:500 |

[a]BA-D5 and SC-71 were deposited to the Developmental Studies Hybridoma Bank (DSHB) by Schiaffino, S. [b]2E8 was deposited to the DSHB by Engvall, E.S. *SC-71 stains both IIA and IIX fibres in human tissue.

## Differential expression analysis

Differential expression was done using *FindMarkers* from the Seurat (v4), comparing the gene expression between the groups, using the non-parametric Wilcoxon rank sum test. *P* values were Bonferroni corrected using all features in the dataset (Hao et al., 2021).

Functional analyses were done using the 'DAVID 2021 (Dec. 2021), DAVID Knowledgebase (v2023q4, updated quarterly)' (Huang et al., 2008; Sherman et al., 2022). Genes identified as significantly regulated (adjusted *p*-value <0.05; i.e. differentially expressed genes [DEGs]) from the analysis of the myonuclei subclusters were uploaded as gene lists to the DAVID platform and analysed using the KEGG pathway tool. The resulting lists were then reduced to only include pathways with a *P* value <0.05.

## Immunofluorescence, microscopy and image analysis

The Tissue-Tek embedded portion of the muscle biopsy was cut in the cross-sectional plane of the muscle fibres in 10 μm thick sections in a cryostat at −20°C, placed on glass slides, and allowed to dry at room temperature before being stored at −80°C.

For the immunofluorescent staining, slides were removed from the freezer and dried for 20 min at room temperature. Between each step in the protocol, slides were washed 3 × 5 min in 1× PBS. All primary and secondary antibodies were diluted in 1% bovine serum albumin (BSA, IgG free) in 1× PBS (a list of antibodies is provided in Table 1). The staining protocol was performed over 2 days as follows: Day 1, fixation for 5 min in 4% paraformaldehyde, followed by overnight incubation with primary antibodies at 4°C. Day 2, incubation with secondary antibodies for 45 min at room temperature, followed by mounting with cover glasses in ProLong Gold (Molecular Probes ProLong Gold anti-fade reagent, cat. no. P36931). Slides were dried in the dark at room temperature for 2 days and then stored at −20°C until imaging.

Four-channel images covering all fibres in the biopsy section were taken and automatically stitched together with a Zeiss Axioscan Z1 fluorescent microscope, using a 10× objective. Type I and type II fibres were subsequently counted manually in ImageJ (version 1.53c; National Institutes of Health, Bethesda, MD, USA), using the ObjectJ plugin. The analysis was performed by an investigator who was blinded with respect to the biopsy ID.

## Statistical analyses

Normal distribution of fibre type distribution (% type I fibres) in the immunofluorescence analysis was tested by Shapiro–Wilk test and inspection of qq-plots, and two-way repeated measures ANOVA was used for comparing between the groups and training state. The squared Pearson's coefficient and corresponding *P* value were calculated for correlations between the proportion of type I fibres in the immunofluorescence analysis and type I myonuclei (out of the total sum of type I, IIA and IIX myonuclei) in the snRNA-seq dataset. GraphPad Prism (vers.10.2.3) was used, and *P* values <0.05 were considered significant.

## Results

Participant characteristics have previously been published (Dela et al., 2019). In brief, the participants with type 2 diabetes were 57 ± 2 years old, with a BMI of 31 ± 1 kg/m$^2$, glycated haemoglobin (HbA1c) levels of 53 ± 1 mmol/mol, and a $\dot{V}_{\text{O}_2\text{peak}}$ of 29.3 ± 1.2 ml/min/kg. The healthy controls were 53 ± 2 years old, with a BMI of 31 ± 1 kg/m$^2$, HbA1c levels of 37 ± 1 mmol/mol, and a $\dot{V}_{\text{O}_2\text{peak}}$ of 31.5 ± 1.2 ml/min/kg. After 2 weeks of one-legged HIIT, insulin-stimulated glucose clearance in the trained leg was ∼30% higher compared with the

untrained leg in both groups (stage 1: 33% and stage 2: 24% in the diabetes group; stage 1: 31% and stage 2: 29% in the controls).

## snRNA-seq clustering and annotation

An overview of the experimental workflow is shown in Fig. 1. One participant in the diabetes group did not have enough tissue for snRNA-seq, and nuclear transcriptomes were profiled in 38 muscle biopsies (controls $n = 10$; type 2 diabetes $n = 9$) and subjected to bioinformatical analysis. After the removal of doublets, a total of 135,225 nuclei remained for further analysis (average per sample: 3558 (range 1784–4716) nuclei), with an average of 2198 (range 1779–2631) genes per nucleus.

Subsequent integration of all nuclei transcriptional profiles followed by unbiased clustering resulted in the two myonuclear clusters slow muscle cells (*MYH7, PPARGC1A*) and fast muscle cells (*MYH1, MYH2, CKM*); and nine mononuclear cell clusters defined by their expression of the following markers: satellite cells (*PAX7*), fibroblasts (*PDGFRA, THY1*), adipocytes (*ADIPOQ*), Schwann cells (*MBP, PLP1, MPZ*), immune cells (*CD163,*

*PTPRC*), smooth muscle cells (*ACTA2, MYH11*) and endothelial cells (*VWF, PECAM1*). The latter was subdivided into artery cells (*SOX17*), vein cells (*SELP, ACKR1*) and capillary cells (*LPL*) (Figs 2*A* and 3) (Kedlian et al., 2024; Lai et al., 2024; Massier et al., 2023; Pass et al., 2023; Perez et al., 2022; Soule et al., 2023). The proportion of the different cell types and myonuclei within each biopsy is shown in Fig. 2*B*. Except for Schwann cells, all cell types were present in all samples (Fig. 3).

Next, all skeletal muscle nuclei clusters were subjected to subcluster analysis (Fig. 4), resulting in the segregation into the three known major fibre types (type I, type IIA and type IIX) (Lai et al., 2024; Pass et al., 2023) and four other myocyte subclusters. Of these, one expressed a marker for the myotendinous junction (*COL22A1*), one expressed markers previously associated with senescence (*LRRK2, CDNKA1*) (Perez et al., 2022) and two clusters (unknown_1 and unknown_2) did not show a clear expression profile. One sample from a subject in the diabetes group consisted predominantly of nuclei belonging to the unknown_2 cluster, and both samples from this subject were therefore removed before performing the subsequent differential gene expression analysis (controls $n = 10$, type 2 diabetes $n = 8$). For the

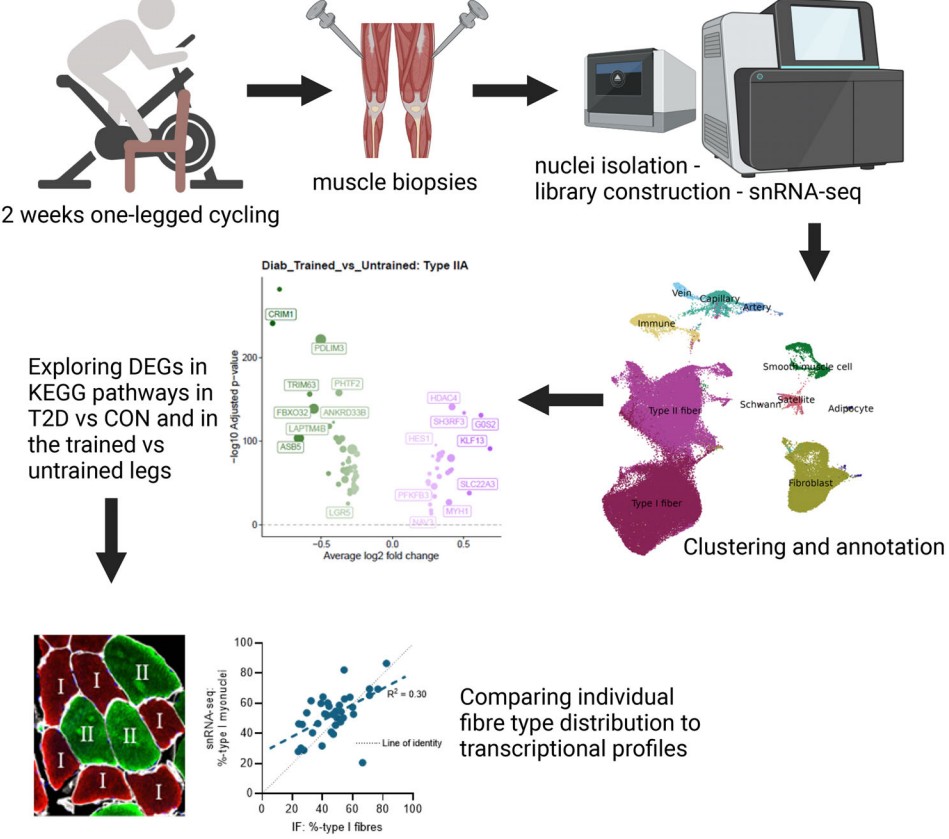

**Figure 1. An overview of the workflow**
Created in BioRender. Hansen, M. (2025) https://BioRender.com/q05o255. [Colour figure can be viewed at wileyonlinelibrary.com]

three major fibre types, the data analysis included a total of 47,662 type I myonuclei (range 151–2545), 35,447 type IIA myonuclei (range 145–1711) and 8331 type IIX myonuclei (range 3–728).

### Fibre type assessment by immunofluorescence and snRNA-seq

To assess if the muscle fibre type composition revealed by snRNA-seq is on par with more traditional methods, we stained histological specimens from the 18 participants included in the DEG analysis using immunofluorescence. A representative image of a muscle biopsy cross-section stained with antibodies for type I myosin and laminin

is shown in Fig. 5*A*. The immunofluorescence analysis of the proportion of type I fibres as a percentage of the total number of fibres for each biopsy included 494 ± 221 fibres per sample (range 133–1040). There was no main effect of type 2 diabetes or training on fibre type distribution (%type I fibres), though significant individual variability was observed ($P = 0.0209$) (Fig. 5*B*). The proportion of type I fibres in the immunofluorescence analysis correlated with the proportion of type I myonuclei identified in the snRNA-seq dataset ($R^2 = 0.30$, $P = 0.0005$, Fig. 5*C*). For comparison the total count for each fibre type measured by immunofluorescence and snRNA-seq is shown in Table 2. In the control group, HIIT resulted in a higher prevalence (average log2 fold change of 0.339,

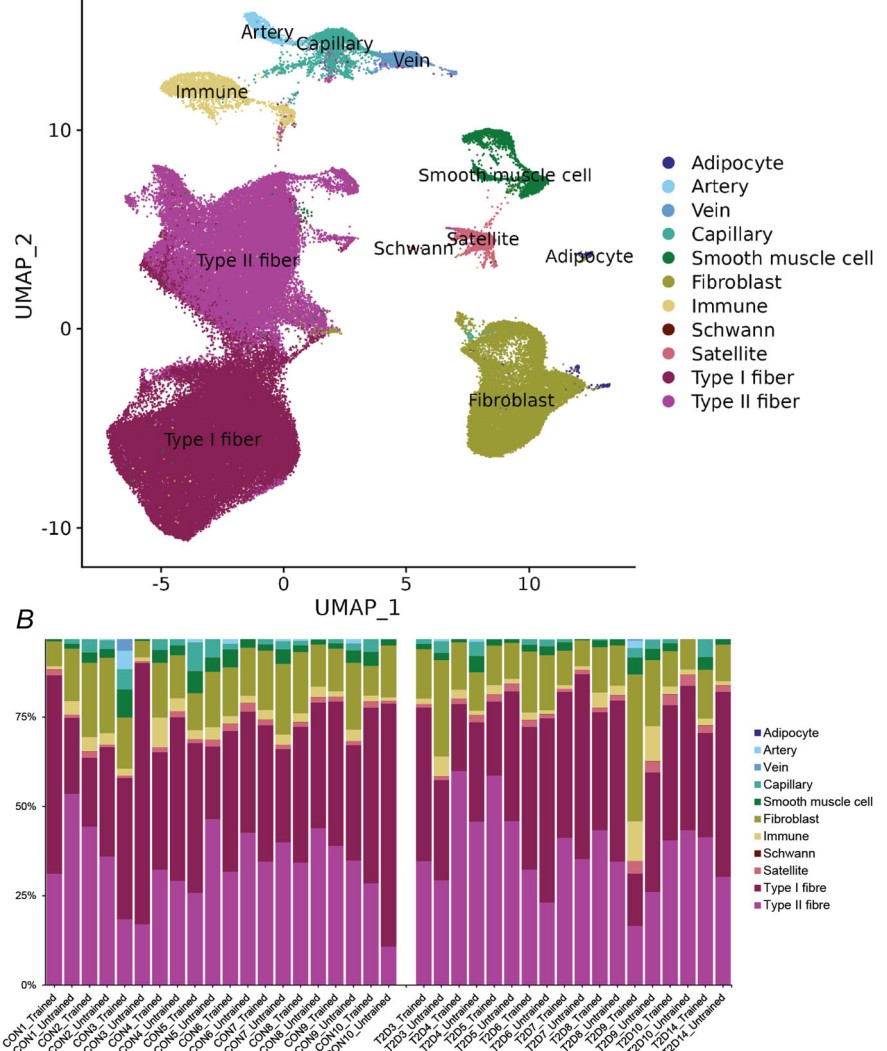

**Figure 2. Cell type distribution**
*A*, UMAP of the skeletal muscle cell clusters. *B*, the proportions of the different cell types in each muscle sample.
[Colour figure can be viewed at wileyonlinelibrary.com]

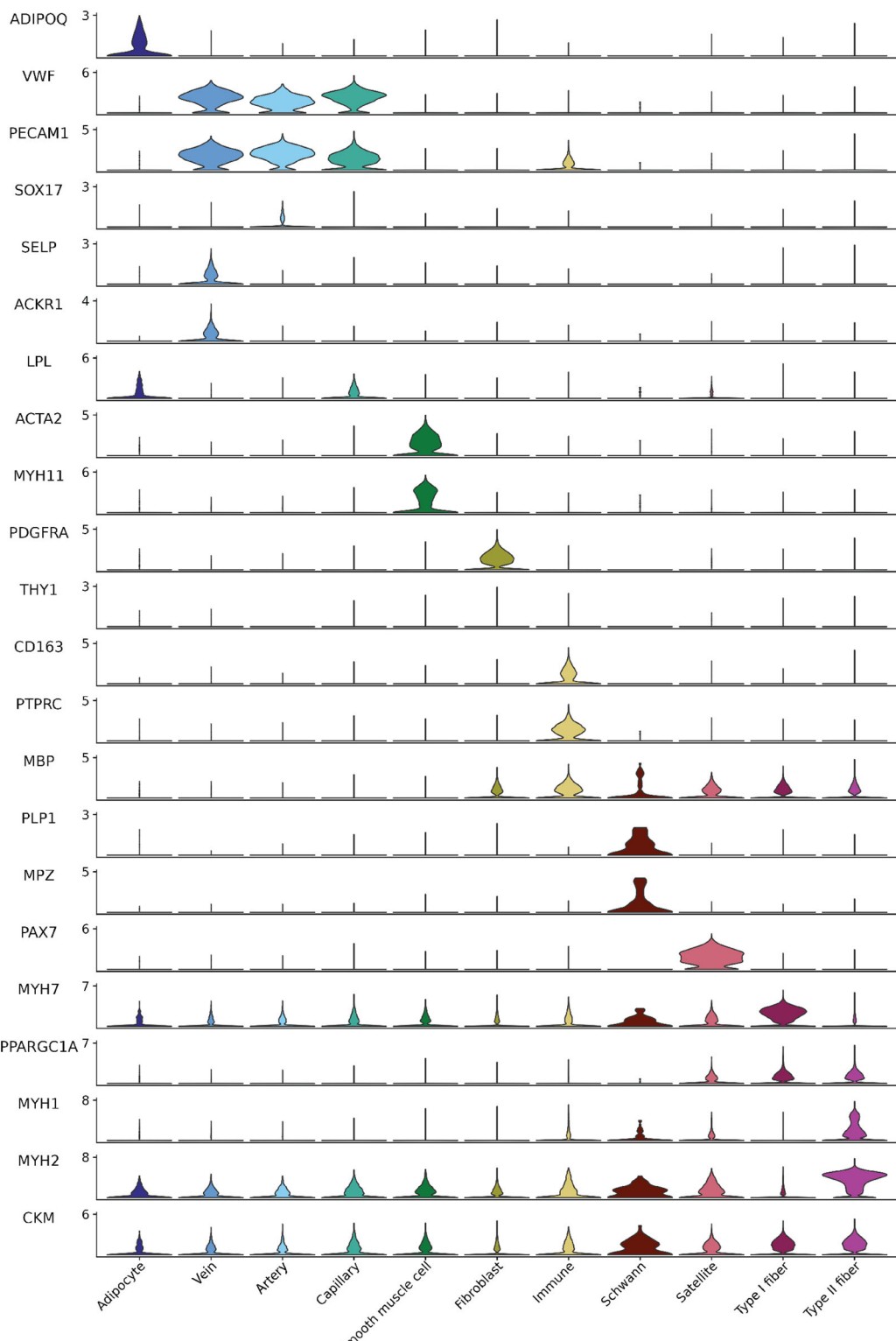

**Figure 3. Stacked violin plots of canonical marker gene expression across the 11 snRNA-seq clusters used for cell type identification**
[Colour figure can be viewed at wileyonlinelibrary.com]

$P = 4.8 \times 10^{-8}$) of *MYH2* expression (type IIA myosin) in type IIX (*MYH1$^+$*) myonuclei (Fig. 5*D*).

### Differentially expressed genes (DEGs) and Kyoto Encyclopedia of Genes and Genomes (KEGG) pathways

We identified 249 DEGs in type I/IIA/IIX myonuclei (95 with relative higher expression in the controls and 154 with relative higher expression in type 2 diabetes) (Fig. 6*A*)

when comparing nuclear RNA expression between the untrained legs of the control and type 2 diabetes groups. This indicates a difference in the transcriptional profile between the control and diabetes groups at the level of the muscle fibre. The effect of HIIT was assessed by comparing the trained leg with the untrained leg within each group (Fig. 6*B* and *C*), revealing similar numbers of DEGs in controls (222 DEGs; 135 with relative lower expression and 87 with relative higher expression in the trained compared to the untrained leg) and type 2 diabetes (234 DEGs; 140 with relative lower expression and 94 with

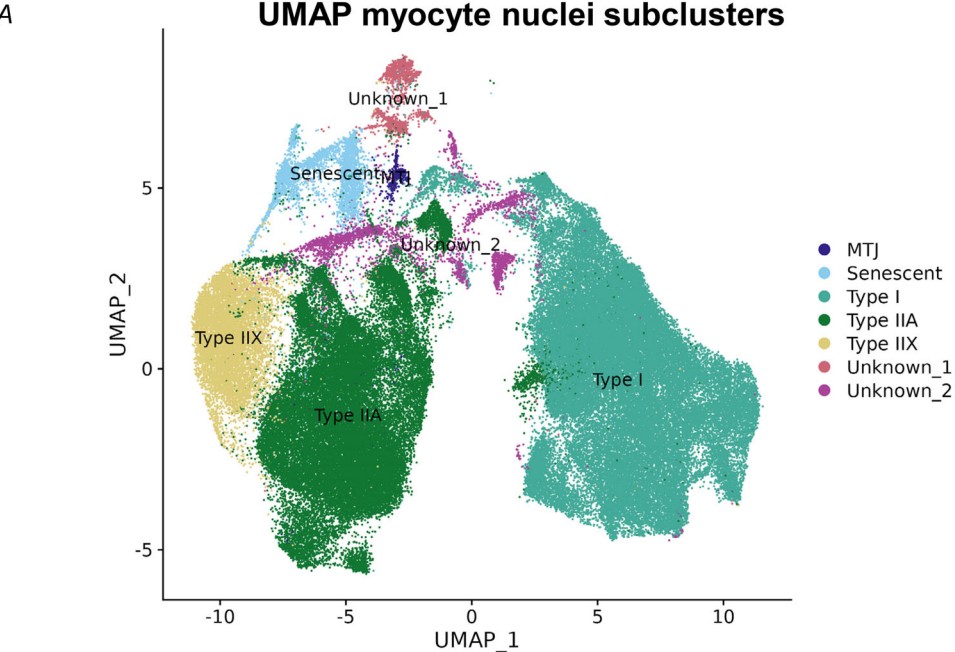

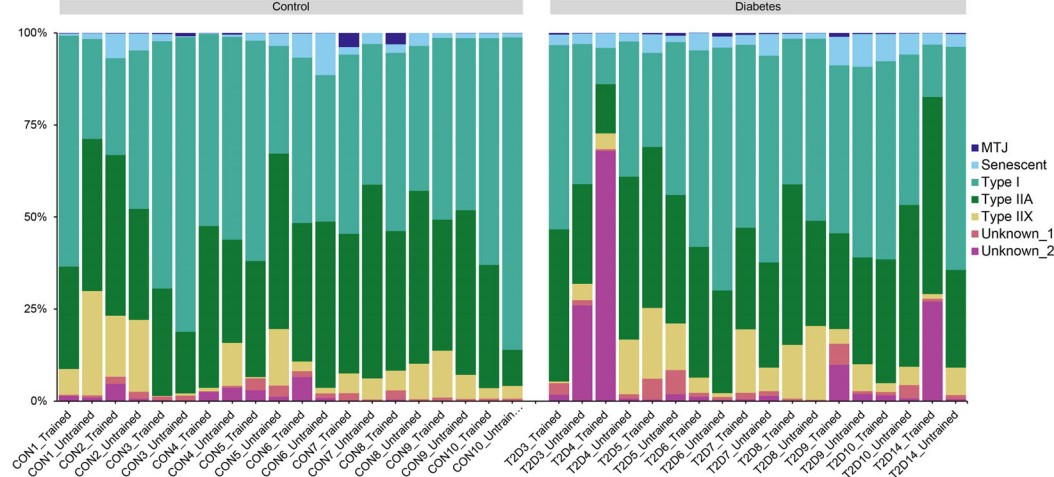

**Figure 4. Myonuclei**
*A*, UMAP of myonuclei subclusters. *B*, The proportions of the different myonuclei in each muscle sample. Abbreviation: MTJ, myotendinous junction. [Colour figure can be viewed at wileyonlinelibrary.com]

relative higher expression in the trained compared to the untrained leg).

To further explore the transcriptional heterogeneity of myonuclei in type 2 diabetes *vs.* controls and its response to HIIT, we searched for significantly regulated KEGG pathways across all fibre types and conditions. The three differentially regulated KEGG pathways with the lowest *P* values are shown for the untrained leg in type 2 diabetes *vs.* controls (Fig. 6*D*), for the effect of HIIT in the controls (trained *vs.* untrained leg, Fig. 6*E*) and for the effect of HIIT in type 2 diabetes (trained *vs.* untrained leg, Fig. 6*F*). For the full list of KEGG-pathways we refer to Table 3.

In the untrained leg, the top three identified KEGG pathways in type I and IIA myonuclei included insulin resistance (hsa04931) and insulin signalling pathway (hsa04910) (Fig. 6*D*). In response to HIIT in the control group, there were no significantly regulated KEGG pathways in the type I myonuclei, while the top three regulated KEGG pathways in

the type IIA myonuclei included glucagon signalling (hsa04022) and Insulin resistance as well as the metabolic glycolysis/gluconeogenesis pathway (hsa00010, Fig. 6*E*). These pathways were not among the top three most differentially expressed KEGG pathways in response to HIIT in the diabetes group (Fig. 6*F*). In both controls and type 2 diabetes, HIIT resulted in differentially expressed cardiomyopathy-related KEGG pathways.

## Effect of type 2 diabetes and HIIT on genes in major metabolic pathways

Impaired insulin sensitivity and glucose uptake capacity are pathophysiological/phenotypic hallmarks of type 2 diabetes. Therefore, we proceeded with a graphical presentation of the DEGs in type I, IIA and IIX myonuclei involved in relevant major metabolic pathways (Fig. 7*A* and *B*). Secondly, we included

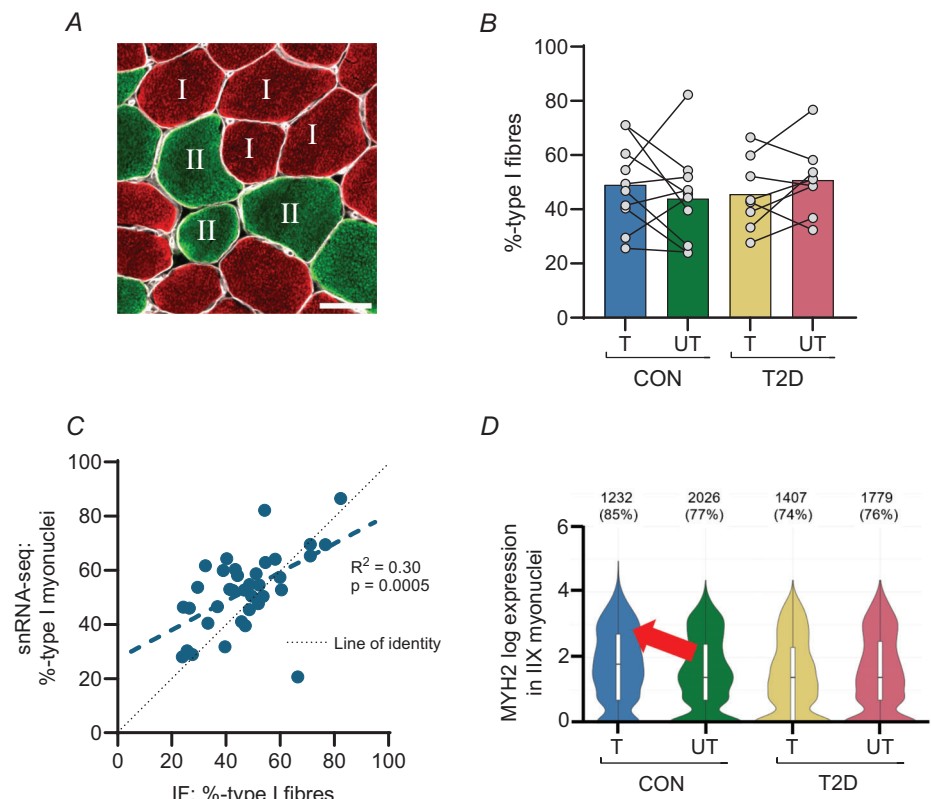

**Figure 5. Immunofluorescence analysis and snRNA-seq assessment of myofibre type composition and training response in control and type 2 diabetes muscle**
*A*, a representative image of immunostaining of a muscle biopsy cross-section, with type I and type II fibres denoted. Scale bar = 50 μm. *B*, type I myofibre proportion observed by immunofluorescence (IF) analysis in the trained and untrained legs of the control group (CON) and the type 2 diabetes (T2D) group. *C*, correlation between the proportion of type I fibres from the IF analysis and the proportion of type I myonuclei (% of the sum of type I, IIA and IIX myonuclei) from the snRNA-seq. *D*, the expression of MYH2 (type IIA myosin) in type IIX (MYH1+) myonuclei in the trained and untrained legs of CON and T2D. Box plots are shown as median, 25th and 75th percentiles. The number and percentage of cells expressing MYH2 are printed above the respective condition. The red arrow denotes a significant difference between the trained and untrained legs in CON. [Colour figure can be viewed at wileyonlinelibrary.com]

**Table 2. Number of fibres in the immunofluorescence analysis and number of myonuclei in the snRNA-seq analysis per fibre type**

| | Immunofluorescence (fibres) | | | snRNA-seq (nuclei) | | |
| | Type I | Type II | All (Type I+II) | Type I | Type II[a] | All (Type I+II) |
| --- | --- | --- | --- | --- | --- | --- |
| **Sum** | 8399 | 9398 | 17,797 | 46,418 | 41,745 | 88,163 |
| **Average per biopsy (range)** | 233 (49–554) | 261 (62–607) | 494 (133–1040) | 1289 (254–2545) | 1160 (167–2064) | 2449 (421–3555) |
| **Number of biopsies** | 36 | | | 36 | | |

[a]Type II myonuclei include the type IIA and IIX myonuclei from the snRNA-seq analysis. The immunofluorescence analysis consisted only of the distinction between type I and II myofibres.

a graphical presentation of the DEGs in the two differentially expressed KEGG-pathways related to insulin sensitivity (hsa04910:insulin-signalling and hsa04931:insulin-resistance) (Fig. 7C and D) (Kanehisa Laboratories, 2021, 2023). In both cases, two conditions were examined for DEGs; (1) the effect of type 2 diabetes in untrained muscle (diabetes *vs.* controls in the untrained leg (Fig. 7A and C)); and (2) the response to HIIT in controls and diabetes (trained *vs.* untrained leg within each group (Fig. 7B and D)).

Glucose-related metabolic pathways of interest were glycogenolysis, glycolysis, gluconeogenesis and the citric acid cycle. Fatty acid-related metabolic pathways included the fatty acid to acyl-CoA conversion, the carnitine shuttle and beta-oxidation. Except for ACAT1 (beta-oxidation), DEGs were only present in the glucose-related metabolic pathways (glycogenolysis and glycolysis/gluconeogenesis, Fig. 7A and B). There were very few differences between the diabetes and control group in the untrained leg, where *ACAT1* and *PYGM* were relatively more highly expressed in diabetes in type I and type IIX fibres, respectively (Fig. 7A). In response to HIIT, the control group demonstrated a concerted induction of nuclear RNA specifically related to glycogen breakdown and glycolysis (*PYGM, PGM1, ENO3, GPI, PKM, AGL*), which was mainly driven by the type IIA fibres and to some extent the IIX fibres (Fig. 7B). This regulation was non-significant in the diabetes group. An overview of the relative expression levels of the seven genes, along with an overview of their positions in the glycogenolysis and glycolysis pathways, is shown in Fig. 8.

### Effect of type 2 diabetes and HIIT on genes in the KEGG insulin resistance and insulin signalling pathways

The two examined KEGG-pathways included a total of 189 genes, of which 56 were assigned to both pathways

(Fig. 7E). In the untrained leg, nine DEGs (Fig. 7C) were differentially expressed in one or more of the type I (six DEGs), IIA (four DEGs) and IIX (three DEGs) myonuclei. *PPP1R3C* was the only DEG that appeared in all three fibre types and was consistently relatively more highly expressed in the untrained leg in the controls compared to the untrained leg in type 2 diabetes. In contrast, *PPARGC1A* was relatively more highly expressed in type IIA myonuclei, but relatively less expressed in type I myonuclei in the untrained leg in the controls compared to type 2 diabetes.

In summary, the response to HIIT in the controls was predominantly seen in type IIA myonuclei (seven DEGs) and, to a lesser extent, the type IIX myonuclei (two DEGs, Fig. 7D). In contrast, the response to HIIT in type 2 diabetes was distributed across all myonuclei with three DEGs in type I, two DEGs in type IIA and one DEG in type IIX (Fig. 7D).

### Discussion

We performed snRNA-sequencing transcriptomics on muscle biopsies from individuals with and without type 2 diabetes after 2 weeks of one-legged HIIT on a bicycle ergometer (i.e. chronic exercise and not just the effect of a single exercise bout). We confirmed the fibre type distribution with immunofluorescence, which correlated with the distribution of the myonuclei. We focused our further analysis on the most significantly regulated parts of the transcriptome related to metabolic pathways and the insulin resistance/insulin signalling KEGG pathways and were surprised that we found only a few DEGs comparing relative expression between the two groups in the untrained leg. Interestingly, when examining the metabolic pathways, only the non-diabetic group showed a substantial transcriptional response to HIIT in the trained leg, and primarily in the type IIA fibres.

## Immunofluorescence fibre type distribution confirms the distribution of snRNA-seq myonuclei

Type I muscle fibres have been proposed to play a larger role in glucose homeostasis than type II fibres (Frankenberg et al., 2022), but it is unclear whether individuals with type 2 diabetes have a different muscle fibre type distribution compared to healthy individuals (Frankenberg et al., 2022; Gaster et al., 2001; He et al., 2001; Mårin et al., 1994; Oberbach et al., 2006; Vind et al., 2012). In the present study, we found no differences between the groups with and without type 2 diabetes. We found a significant correlation between the proportion of type I myonuclei identified in the snRNA-seq data and the type I myofibres observed with immunofluorescence, indicating that snRNA-seq analysis can be used to estimate fibre type distribution in middle-aged sedentary individuals with and without type 2 diabetes.

Training-induced fibre-type switching has been shown related to different exercise modalities (Plotkin et al., 2021). The SC-71 antibody used in the immuno-fluorescence analysis stains both type IIA and IIX fibres in human tissue, but at the transcriptional level, we did see some indications of transformation in the type II

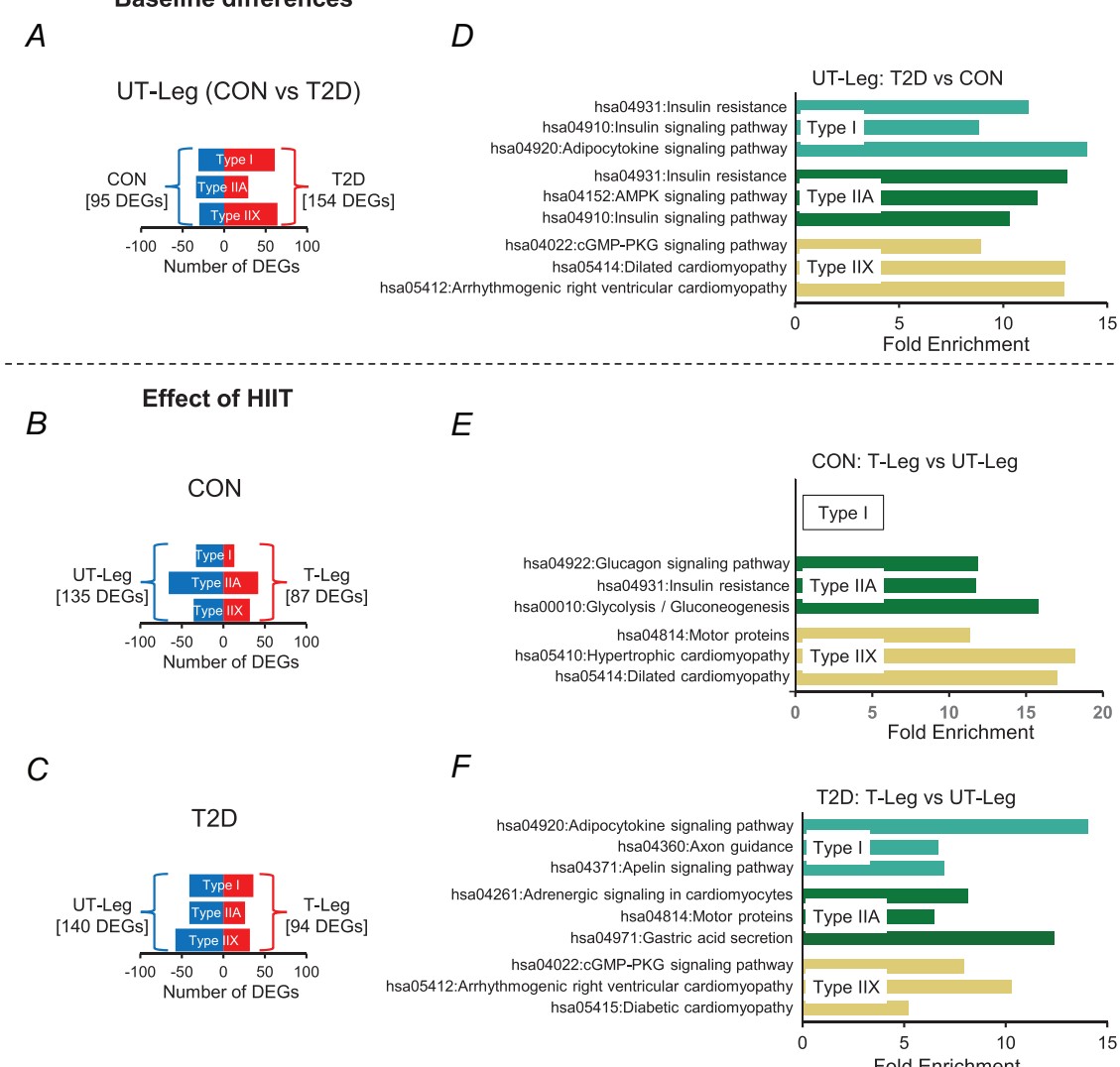

**Figure 6.  Differentially expressed genes (DEGs) and Kyoto Encyclopedia of Genes and Genomes (KEGG) pathways**
*A–C*, number of differentially expressed genes (DEGs) in the untrained (UT) legs between the control group (CON) and the diabetes group (T2D) (*A*), and between the trained (T) and untrained legs of CON (*B*) and T2D (*C*). *D–F*, the top three differentially expressed KEGG pathways in the untrained legs between T2D and CON (*D*), and between the trained and untrained legs of CON (*E*) and T2D (*F*). There were no significantly regulated KEGG pathways between the trained and untrained legs in the type I myonuclei of the control group. [Colour figure can be viewed at wileyonlinelibrary.com]

**Table 3. Full list of differentially expressed KEGG pathways**

| Category | Term | Count | % | *P* value |
|---|---|---|---|---|
| Baseline differences: Type 2 diabetes *vs.* control | | | | |
| **Type I** | | | | |
| KEGG_PATHWAY | hsa04931:Insulin resistance | 5 | 7.24638 | 8.65E-04 |
| KEGG_PATHWAY | hsa04910:Insulin signalling pathway | 5 | 7.24638 | 0.00209 |
| KEGG_PATHWAY | hsa04920:Adipocytokine signalling pathway | 4 | 5.7971 | 0.00259 |
| KEGG_PATHWAY | hsa04152:AMPK signalling pathway | 4 | 5.7971 | 0.01235 |
| KEGG_PATHWAY | hsa04068:FoxO signalling pathway | 4 | 5.7971 | 0.01528 |
| KEGG_PATHWAY | hsa04211:Longevity regulating pathway | 3 | 4.34783 | 0.04934 |
| **Type IIA** | | | | |
| KEGG_PATHWAY | hsa04931:Insulin resistance | 4 | 8.51064 | 0.00297 |
| KEGG_PATHWAY | hsa04152:AMPK signalling pathway | 4 | 8.51064 | 0.00409 |
| KEGG_PATHWAY | hsa04910:Insulin signalling pathway | 4 | 8.51064 | 0.00579 |
| KEGG_PATHWAY | hsa04936:Alcoholic liver disease | 4 | 8.51064 | 0.0064 |
| KEGG_PATHWAY | hsa04920:Adipocytokine signalling pathway | 3 | 6.38298 | 0.01484 |
| KEGG_PATHWAY | hsa05410:Hypertrophic cardiomyopathy | 3 | 6.38298 | 0.02447 |
| KEGG_PATHWAY | hsa04922:Glucagon signalling pathway | 3 | 6.38298 | 0.0337 |
| **Type IIX** | | | | |
| KEGG_PATHWAY | hsa04022:cGMP-PKG signalling pathway | 6 | 8.33333 | 4.27E-04 |
| KEGG_PATHWAY | hsa05414:Dilated cardiomyopathy | 5 | 6.94444 | 4.94E-04 |
| KEGG_PATHWAY | hsa05412:Arrhythmogenic right ventricular cardiomyopathy | 4 | 5.55556 | 0.00325 |
| KEGG_PATHWAY | hsa05410:Hypertrophic cardiomyopathy | 4 | 5.55556 | 0.00504 |
| KEGG_PATHWAY | hsa04020:Calcium signalling pathway | 5 | 6.94444 | 0.01614 |
| KEGG_PATHWAY | hsa04971:Gastric acid secretion | 3 | 4.16667 | 0.03511 |
| **Control: Trained *vs.* untrained** | | | | |
| **Type I** | | | | |
| N/A | | | | |
| **Type IIA** | | | | |
| KEGG_PATHWAY | hsa04922:Glucagon signalling pathway | 6 | 7.5 | 1.20E-04 |
| KEGG_PATHWAY | hsa04931:Insulin resistance | 6 | 7.5 | 1.26E-04 |
| KEGG_PATHWAY | hsa00010:Glycolysis/Gluconeogenesis | 5 | 6.25 | 2.39E-04 |
| KEGG_PATHWAY | hsa04910:Insulin signalling pathway | 6 | 7.5 | 3.82E-04 |
| KEGG_PATHWAY | hsa00500:Starch and sucrose metabolism | 4 | 5 | 5.81E-04 |
| KEGG_PATHWAY | hsa05410:Hypertrophic cardiomyopathy | 5 | 6.25 | 7.40E-04 |
| KEGG_PATHWAY | hsa00030:Pentose phosphate pathway | 3 | 3.75 | 0.008294 |
| KEGG_PATHWAY | hsa05414:Dilated cardiomyopathy | 4 | 5 | 0.009615 |
| KEGG_PATHWAY | hsa04814:Motor proteins | 5 | 6.25 | 0.011555 |
| KEGG_PATHWAY | hsa01200:Carbon metabolism | 4 | 5 | 0.01565 |
| KEGG_PATHWAY | hsa04152:AMPK signalling pathway | 4 | 5 | 0.017912 |
| KEGG_PATHWAY | hsa04371:Apelin signalling pathway | 4 | 5 | 0.025744 |
| KEGG_PATHWAY | hsa04920:Adipocytokine signalling pathway | 3 | 3.75 | 0.03998 |
| **Type IIX** | | | | |
| KEGG_PATHWAY | hsa04814:Motor proteins | 8 | 14.54545 | 3.75E-06 |
| KEGG_PATHWAY | hsa05410:Hypertrophic cardiomyopathy | 6 | 10.90909 | 1.40E-05 |
| KEGG_PATHWAY | hsa05414:Dilated cardiomyopathy | 6 | 10.90909 | 1.92E-05 |
| KEGG_PATHWAY | hsa04260:Cardiac muscle contraction | 5 | 9.090909 | 2.33E-04 |
| KEGG_PATHWAY | hsa04261:Adrenergic signalling in cardiomyocytes | 5 | 9.090909 | 0.001997 |
| KEGG_PATHWAY | hsa04919:Thyroid hormone signalling pathway | 4 | 7.272727 | 0.008731 |
| KEGG_PATHWAY | hsa04020:Calcium signalling pathway | 5 | 9.090909 | 0.011587 |
| KEGG_PATHWAY | hsa04022:cGMP-PKG signalling pathway | 4 | 7.272727 | 0.020724 |
| KEGG_PATHWAY | hsa05415:Diabetic cardiomyopathy | 4 | 7.272727 | 0.0343 |
| KEGG_PATHWAY | hsa04970:Salivary secretion | 3 | 5.454545 | 0.042565 |

*(Continued)*

**Table 3. (Continued)**

| Category | Term | Count | % | P value |
|---|---|---|---|---|
| **Type 2 Diabetes: Trained vs. untrained** | | | | |
| **Type I** | | | | |
| KEGG_PATHWAY | hsa04920:Adipocytokine signalling pathway | 4 | 5.797101 | 0.002589 |
| KEGG_PATHWAY | hsa04360:Axon guidance | 5 | 7.246377 | 0.005793 |
| KEGG_PATHWAY | hsa04371:Apelin signalling pathway | 4 | 5.797101 | 0.017883 |
| **Type IIA** | | | | |
| KEGG_PATHWAY | hsa04261:Adrenergic signalling in cardiomyocytes | 4 | 6.779661 | 0.011297 |
| KEGG_PATHWAY | hsa04814:Motor proteins | 4 | 6.779661 | 0.020631 |
| KEGG_PATHWAY | hsa04971:Gastric acid secretion | 3 | 5.084746 | 0.022495 |
| KEGG_PATHWAY | hsa05412:Arrhythmogenic right ventricular cardiomyopathy | 3 | 5.084746 | 0.023051 |
| KEGG_PATHWAY | hsa04260:Cardiac muscle contraction | 3 | 5.084746 | 0.028926 |
| KEGG_PATHWAY | hsa05410:Hypertrophic cardiomyopathy | 3 | 5.084746 | 0.030795 |
| KEGG_PATHWAY | hsa04970:Salivary secretion | 3 | 5.084746 | 0.03271 |
| KEGG_PATHWAY | hsa05414:Dilated cardiomyopathy | 3 | 5.084746 | 0.034672 |
| **Type IIX** | | | | |
| KEGG_PATHWAY | hsa04022:cGMP-PKG signalling pathway | 5 | 6.944444 | 0.003034 |
| KEGG_PATHWAY | hsa05412:Arrhythmogenic right ventricular cardiomyopathy | 3 | 4.166667 | 0.032035 |
| KEGG_PATHWAY | hsa05415:Diabetic cardiomyopathy | 4 | 5.555556 | 0.037323 |

myonuclei in CON, possibly towards a more oxidative profile with more *MYH2* (encoding myosin heavy chain 2, a marker of type IIA myofibres) in the type IIX (*MYH1*[+]) myonuclei in the trained leg. This was not parallelled in the diabetes group. In addition, KEGG pathways related to cardiomyopathy were among the top three differentially expressed pathways in IIX myonuclei in both groups. This was caused by differential expression of genes encoding for proteins well described in the skeletal muscle, such as myofibrillar/cytoskeletal proteins (*ACTN2*, *ACTN3*, *TPM1*, *TPM3*, *TTN*, *MYH7*) and the Serca pumps (*ATP2A1*, *ATP2A2*). Rather than any true relation to the heart or cardiac function, these changes possibly reflected ongoing skeletal muscle remodelling processes in response to HIIT.

## Transcriptional differences between the type 2 diabetes and control group in the untrained leg

In skeletal muscle, we found 249 DEGs in the type I, type IIA and type IIX myonuclei between the untrained legs of the control and diabetes group. In comparison, a study using bulk RNA-sequencing of muscle samples from severely insulin-resistant, insulin-treated individuals with type 2 diabetes and healthy controls reported 117 DEGs (Møller et al., 2017). There was very little overlap between the DEGs in these two datasets, which may be explained by methodological differences, as bulk RNA-seq contains mRNA from all cell types in the muscle and does not distinguish between slow and fast muscle fibres. Additionally, the two methods are not examining the same

pool of RNA, and for adipose tissue Massier et al. (2023) have shown that snRNA-seq does not fully correlate with RNA-seq. In contrast, the snRNA-seq method allowed us to perform high-resolution transcriptional profiling of the individual myonuclei, resulting in the discovery of DEGs that may not be detected with bulk RNA-seq. The discrepancies between the DEGs reported by Møller et al. (2017) and the present data could also be due to the differences in severity and duration of type 2 diabetes (time since diagnosis: 17 years *vs*. 5 years in our study) and/or the tendency to a higher BMI in the diabetes group (controls: 28 kg/m$^2$; diabetes: 36 kg/m$^2$; $P = 0.05$) in their study (Møller et al., 2017). The two groups in our study were well-matched on many different parameters, including age, BMI and whole-body fat percentage (Dela et al., 2019). In contrast to Møller's and our findings, bulk RNA-sequencing in a large number of human skeletal muscle samples from individuals with normal glucose tolerance ($n = 97$) and type 2 diabetes ($n = 67$) revealed surprisingly few differentially expressed genes (three DEGs in total) (Scott et al., 2016).

In the present study, we found only two DEGs related to major metabolic pathways when comparing transcriptional profiles of myonuclei in the untrained leg in the control *vs*. diabetes group. In contrast, a meta-analysis of six microarray datasets from skeletal muscle showed distinct downregulation of 14 metabolic pathways in type 2 diabetes, including glycolysis/gluconeogenesis and beta-oxidation pathways (Väremo et al., 2015). Though the method and the data presentation differ, the meta-analysis results suggest a larger effect of the type 2 diabetic condition on the metabolic pathways

**Differentially expressed genes in metabolic pathways in type I, IIA, IIX myonuclei**

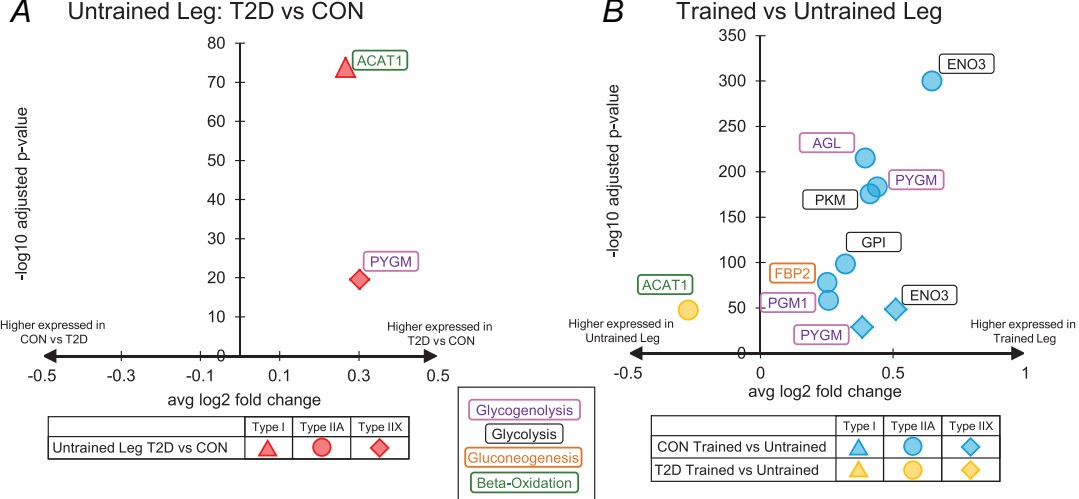

**Differentially expressed genes in insulin related KEGG-pathways in type I, IIA, IIX myonuclei**

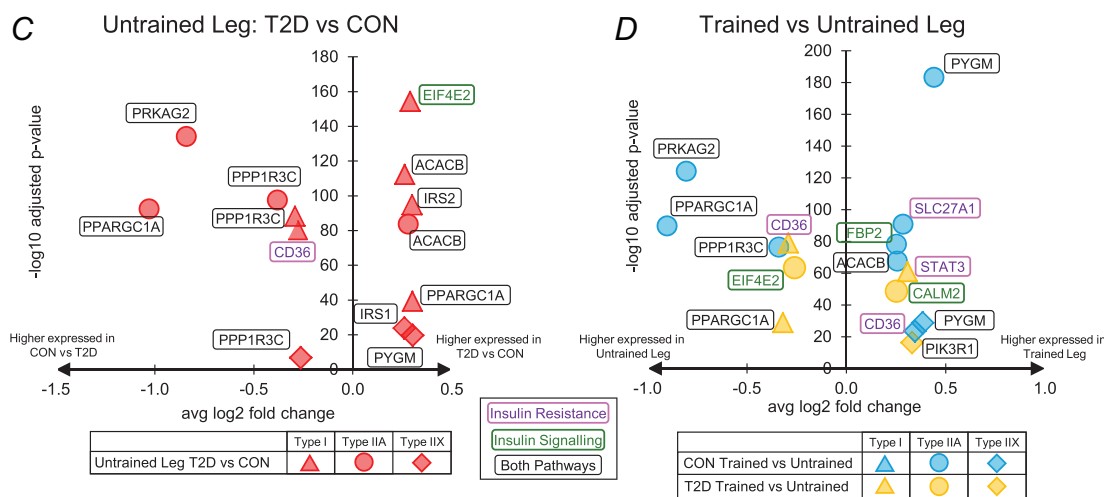

*E* **Number of genes in insulin related KEGG-pathways**

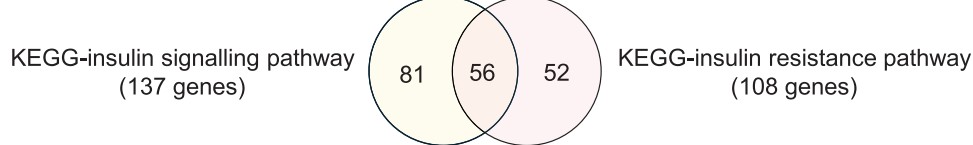

**Figure 7. Differentially expressed genes in metabolic pathways in type I, IIA, and IIX myonuclei**
Differentially expressed genes (DEGs) related to metabolic pathways between CON and T2D in the untrained legs (*A*) and between the trained and untrained legs within the groups (*B*). Additionally, DEGs related to insulin-related KEGG pathways between CON and T2D in the untrained legs (*C*) and between the trained and untrained legs within the groups (*D*). *E*, a Venn diagram of all genes related to the KEGG insulin signalling and KEGG insulin resistance pathways. [Colour figure can be viewed at wileyonlinelibrary.com]

than our results do. Interestingly, Väremo et al. originally included eight studies, but because two of them correlated negatively with the other six, those two were excluded to obtain a more homogenous phenotype. The authors did not find muscle type or contamination to be the cause of the discrepancies, and neither were the age, BMI, or fasting glucose of the participants (Väremo et al., 2015). The exclusion of the studies made the genetic profile and results of the meta-analysis clearer and more significant, but valuable information may have been lost. Finding the underlying reason(s) for the discrepancies between the studies may contribute to our understanding of the different results, whether it be due to the methods used or the participant characteristics.

## HIIT-induced metabolic transcription occurs primarily in type IIA myonuclei and is blunted in type 2 diabetes

In the control group, 2 weeks of HIIT induced a significant transcriptional upregulation of genes related to glycolytic and glycogenolytic metabolic pathways, primarily in type IIA myonuclei and to a lesser extent in type IIX myonuclei. The different metabolic characteristics of the fibre types have been known for many years, with the slow oxidative type I and the fast, more glycolytic type II fibres, the latter being subdivided into the oxidative-glycolytic type IIA and the glycolytic type IIX fibres (Blaauw et al., 2013; Saltin & Gollnick, 1983). These characteristics have recently been confirmed by single-fibre proteomics (Deshmukh et al., 2021; Murgia

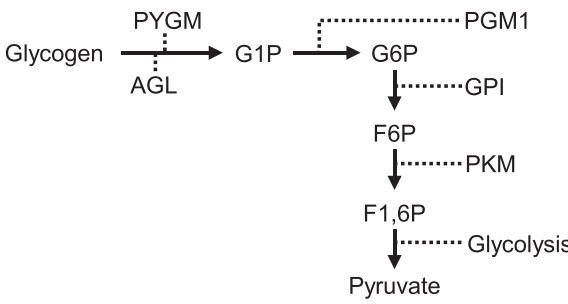

| Gene | Encoded protein | Type I | | | | Type IIA | | | | Type IIX | | | |
|---|---|---|---|---|---|---|---|---|---|---|---|---|---|
| | | CON T | CON UT | T2D T | T2D UT | CON T | CON UT | T2D T | T2D UT | CON T | CON UT | T2D T | T2D UT |
| AGL | Glycogen debranching enzyme | 3.94 | 3.53 | 3.24 | 3.48 | 7.75 | 5.65 | 6.38 | 6.34 | 7.96 | 6.68 | 6.61 | 7.16 |
| ENO3 | Enolase 3 | 0.64 | 0.45 | 0.33 | 0.49 | 2.84 | 1.46 | 1.87 | 1.70 | 2.97 | 1.79 | 1.84 | 2.26 |
| FBP2 | Fructose-Bisphosphatase 2 | 0.24 | 0.21 | 0.19 | 0.22 | 0.74 | 0.47 | 0.50 | 0.47 | 0.69 | 0.50 | 0.64 | 0.69 |
| GPI | Glucose-6-Phosphate Isomerase | 0.85 | 0.67 | 0.92 | 0.75 | 1.67 | 1.14 | 1.51 | 1.40 | 2.03 | 1.74 | 1.65 | 2.01 |
| PGM1 | Phosphogluco-mutase 1 | 0.01 | 0.01 | 0.01 | 0.01 | 0.01 | 0.01 | 0.01 | 0.01 | 0.01 | 0.01 | 0.01 | 0.01 |
| PKM | Pyruvate Kinase M1/2 | 0.32 | 0.25 | 0.23 | 0.29 | 1.20 | 0.65 | 0.89 | 0.74 | 1.13 | 0.84 | 0.81 | 0.88 |
| PYGM | Glycogen Phosphorylase, Muscle Associated | 1.11 | 0.98 | 0.77 | 1.03 | 2.49 | 1.57 | 2.04 | 1.91 | 2.96 | 2.03 | 2.57 | 2.74 |

**Figure 8. Differential expression of genes in type IIA fibres from trained and untrained legs in controls and in patients with Type 2 diabetes, with pathway context**
Relative expression levels of the seven genes that were differentially expressed in type IIA fibres between the trained and untrained legs of the control group (see Fig. 7*B*), along with an overview of their position in the glycogenolysis and glycolysis pathways. Abbreviations: CON, controls; F6P, fructose 6-phosphate; F1,6P, fructose-1,6-bisphosphate; G1P, glucose 1-phosphate; G6P, glucose 6-phosphate; T, trained leg; T2D, type 2 diabetes; UT, untrained leg. [Colour figure can be viewed at wileyonlinelibrary.com]

et al., 2021), and similarly to our study, glycolytic enzymes increased only in type II fibres in response to 12 weeks of endurance training in healthy young men (Deshmukh et al., 2021). Interestingly, only enolase 3 (encoded by the ENO3 gene) was significantly upregulated as a single protein in the proteomics study by Deshmukh et al., and correspondingly, in our study, ENO3 was the most significantly upregulated gene with the highest log2 fold change. Due to the design of their proteomics study, the distinction between type IIA and IIX was not possible. In the present study, we used a HIIT training stimulus, which has been shown to increase the expression of more genes and proteins than other training modalities in both young and old subjects (Robinson et al., 2017). However, it seems that this had little to no effect on the transcriptional activity in the major metabolic pathways in myonuclei in individuals with type 2 diabetes. In line with the present study, a skeletal muscle microarray study in young healthy men reported glucose metabolism and gluconeogenesis to be among the top five significantly enriched pathways after 6 weeks of HIIT (Miyamoto-Mikami et al., 2018). In that study, however, comparisons were not possible between muscle fibre types.

In addition to the lack of transcriptional response related to glycolysis and glycogenolysis between the trained and untrained legs in type 2 diabetes, DEGs related to the KEGG insulin signalling and insulin resistance pathways were distributed across all three fibre types in that group. In contrast, the control group displayed changes only in type II myonuclei. This altered transcriptional adaptation pattern might indicate some form of compensatory mechanism in the diabetes group, but whether type 2 diabetes leads to difficulties appropriately activating the metabolic response in the type II fibres or to a normal initial but non-persistent response is not known. Small, non-significant gene expression changes related to glycogenesis and glycolysis were present in the type IIA myonuclei in the diabetes group (data not shown), suggesting a smaller or perhaps delayed response to HIIT. It is possible that the individuals with type 2 diabetes simply were unable to properly activate their type II fibres to the same extent as their healthy counterparts. As noted in the original publication (Dela et al., 2019), despite similar peak oxygen consumption ($\dot{V}_{O_2peak}$) and similar oxygen consumption during the training sessions in the diabetes and control group, in the diabetes group, the absolute workload during the training sessions was ~72% of the workload in the control group. One might speculate that the lack of transcriptional response in the diabetes group could be due to the lower absolute workload. However, Nordsborg et al. (2010) found that the relative, and not the absolute, workload determined the increase in peroxisome proliferator-activated receptor gamma, coactivator 1α (PGC-1α/PPARGC1A) mRNA in response to acute exercise in trained individuals.

The previously published study using this cohort demonstrated a robust improvement in insulin-stimulated glucose uptake after HIIT in both groups, with similar increases in muscle glucose clearance in the trained legs of the diabetes and control groups (Dela et al., 2019). Additionally, training resulted in enhanced glucose delivery, probably mediated by increased muscle perfusion and capillary recruitment. Given these findings, it is notable that we observed only a limited transcriptional response in key metabolic pathways in the diabetes group, particularly in glycolysis and glycogen breakdown genes, despite all participants showing improved glucose clearance in the trained leg. This suggests that while metabolic function improves with training, transcriptional regulation at the myonuclear level does not fully account for these changes in individuals with type 2 diabetes.

A potential explanation is that post-transcriptional and post-translational mechanisms, such as altered protein stability, enzyme activation and/or metabolite availability, compensate for the lack of transcriptional adaptation in type 2 diabetes. Previous work has indicated that key proteins involved in glucose metabolism, including GLUT4, hexokinase and glycogen synthase, may be upregulated in type 2 diabetic muscle following training independently of significant mRNA changes (Christ-Roberts et al., 2004; Dela et al., 1995). This underscores the need to consider regulatory processes beyond transcription when interpreting exercise responses in insulin-resistant muscle.

An alternative explanation for the apparent lack of transcriptional response in the diabetes group is greater inter-individual variability, which could obscure statistically significant changes. While our differential gene expression analysis did not identify a consistent upregulation of metabolic genes in response to training in type 2 diabetes, subtle or heterogeneous responses across participants cannot be ruled out. This variability could stem from differences in disease duration, medication use or intrinsic differences in muscle fibre composition and recruitment during exercise.

In terms of the transcriptional response to acute exercise and prolonged adaptations to a repeated training stimulus, the timing of the biopsy collection is important. The acute response to a single HIIT session can result in up to 10-fold increases in mRNA expression of certain genes 4 h after a HIIT exercise bout, with a return to baseline levels 24 h later (Perry et al., 2010). This was the case for PGC-1α mRNA in response to knee extensor exercise in healthy participants (Pilegaard et al., 2003) and is supported by a meta-analysis showing upregulation of PPARGC1A (encoding for PGC-1α) in skeletal muscle 2–5 h after acute exercise but a minor downregulation after 20 h with a similar pattern for endurance and resistance training (Amar et al., 2021). In

our study, PPARGC1A was significantly downregulated in the type IIA myonuclei of the control group after the HIIT intervention, whereas in the diabetes group, it was downregulated in the type I myonuclei, again suggesting different response mechanisms to HIIT for individuals with type 2 diabetes. As the biopsies were obtained 40 h after the last exercise bout, we expect to have captured the more chronic effects of HIIT rather than the acute effects. Protein content in the study by Perry et al. (2010) increased more steadily and, in some cases, even increased despite non-significant minor increases in mRNA expression. This serves as a reminder that transcriptional changes do not always directly translate into corresponding changes at the translational or protein level.

### Strengths and limitations

Limitations to the study design have been covered in the original publication (Dela et al., 2019). snRNA-seq enabled us to identify different types of myonuclei; however, due to the multinucleated nature of muscle fibres, the analysis is limited by the inability to determine which nuclei originate from the same muscle fibre. Additionally, we lose transcripts located in the cytoplasm, which are included in both bulk and single-cell RNA-seq. However, bulk RNA-seq does not analyse muscle fibres separately, and even if time is taken to process and purify the muscle sample, it would probably introduce a bias in which genes can be assessed. Single-cell RNA-seq does not allow for analysis of the myonuclei, and the combination of size and the multinucleated nature of the myofibre has complicated the use of this method, leading to non-representative proportions of myofibres in the analysis (Orchard et al., 2021). Lastly, the KEGG pathways, while informative, have built-in limitations such as only annotating pathways that have been previously curated. This was probably what led to the apparent effect of HIIT on cardiomyopathy-related KEGG pathways in our dataset, although the genes included in the pathways are known to be relevant in skeletal muscle function and remodelling.

### Conclusion

In this study, we applied snRNA-seq on skeletal muscle from age- and BMI-matched individuals with and without type 2 diabetes undergoing 2 weeks of one-legged HIIT. Surprisingly, we only observed a modest number of DEGs between the groups in the untrained leg. Despite a marked improvement in insulin sensitivity in the trained leg in both groups, a specific induction in genes involved in glycogen breakdown and glycolysis in type IIA myonuclei was seen only in the control group, while this

response was blunted in the diabetes group. Thus, this study has provided new insights into the nuclear metabolic adaptations to training in individuals with type 2 diabetes, though the mechanisms behind the improvement in insulin sensitivity remain elusive. Additionally, we found a significant correlation between type I fibre proportion by immunofluorescence and type I myonuclei in snRNA-seq analysis, implying that snRNA-seq can be used to assess fibre type distribution.

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

## Additional information

### Data availability statement

All single snRNA-seq data that support the findings of this study have been deposited in ArrayExpress, with the accession code E-MTAB-15009 (link: E-MTAB-15009 < ArrayExpress < BioStudies < EMBL-EBI). https://www.ebi.ac.uk/biostudies/ArrayExpress/studies/E-MTAB-15009?query=E-MTAB-15009. Other data are avaiable from the corresponding author upon resonable request.

### Competing interests

The authors declared the following potential conflicts of interest with respect to research, authorship and/or publication of this article: Julius Elliot Raagaard Grothen, Nikos Sidiropoulos and Thomas Åskov Pedersen are paid employees at Novo Nordisk.

### Author contributions

Conceptualization: T.Å.P., J.W.H., F.D., M.H. and J.E.R.G. Methodology: J.E.R.G. (lead), M.H., A.K., T.Å.P., N.S. and J.M.M. Software: N.S. (lead), J.E.R.G. and T.Å.P. Validation: M.H., A.K., J.E.R.G., T.Å.P. and N.S. Formal analysis: N.S. (lead), M.H., A.K., J.E.R.G., T.Å.P. and J.M.M. Investigation: J.E.R.G. (lead), M.H., A.K., T.Å.P. and N.S. Resources: F.D., T.Å.P., J.E.R.G. and N.S. Data curation: N.S. (lead), M.H., A.K., J.E.R.G. and T.Å.P. Writing – Original draft: M.H. (lead), A.K. and J.E.R.G. Writing – Review & Editing: M.H. (lead), A.K., J.E.R.G., T.Å.P., N.S., J.W.H. and F.D. Visualization: A.K. (lead), M.H., J.E.R.G., T.Å.P., N.S. and J.M.M. Supervision: F.D., T.Å.P. and J.W.H. Project administration: T.Å.P. (lead) and F.D. Funding acquisition: F.D., J.W.H. and T.Å.P. All authors have approved the final version of the manuscript and agree to be accountable for all aspects of the work. All persons designated as authors qualify for authorship, and all those who qualify for authorship are listed.

### Funding

The financial support from The Danish Council for Independent Research | Medical Science, grant no: 0602-02606B, and from the Nordea Foundation is gratefully acknowledged. Novo Nordisk A/S paid for and executed the snRNA-seq with internal resources.

### Acknowledgements

We want to thank all the participants for their contributions and Jeppe Bach, Regitze Kraunsøe, and Norya Durani (all from the Department of Biomedical Sciences, University of Copenhagen), as well as James Ashmore (Zifo) for technical assistance.

### Keywords

glucose metabolism, HIIT, snRNA-seq, training, type 2 diabetes

### Supporting information

Additional supporting information can be found online in the Supporting Information section at the end of the HTML view of the article. Supporting information files available:

**Peer Review History**

