## [Peer Review History · The Journal of Physiology]

The skeletal muscle response to high-intensity training assessed by single-nucleus RNA-sequencing is blunted in individuals with type 2 diabetes

Maria Hansen, Julius Elliot Raagaard Grothen, Anders Karlsen, Jaime Moreno Martinez, Nikos Sidiropoulos, Jørn Wulff Helge, Thomas Åskov Pedersen, and Flemming Dela

DOI: 10.1113/JP288368

Corresponding author(s): *Flemming Dela (fdela@sund.ku.dk)*

Review Timeline:

Submission Date:	17-Dec-2024
Editorial Decision:	17-Jan-2025
Revision Received:	29-Mar-2025
Accepted:	17-Apr-2025

Senior Editor: Paul Greenhaff

Reviewing Editor: Bettina Mittendorfer

Transaction Report:

Dear Dr Dela,

Re: JP-RP-2024-288368 "The skeletal muscle response to high-intensity training assessed by single-nucleus RNA-sequencing is blunted in individuals with type 2 diabetes" by Maria Hansen, Julius Elliot Raagaard Grothen, Anders Karlsen, Jaime Moreno Martinez, Nikos Sidiropoulos, Jørn Wulff Helge, Thomas Åskov Pedersen, and Flemming Dela

Thank you for submitting your manuscript to The Journal of Physiology. It has been assessed by a Reviewing Editor and by 2 expert referees and we are pleased to tell you that it is potentially acceptable for publication following satisfactory major revision.

LANGUAGE EDITING AND SUPPORT FOR PUBLICATION: If you would like help with English language editing, or other article preparation support, Wiley Editing Services offers expert help, including English Language Editing, as well as translation, manuscript formatting, and figure formatting at www.wileyauthors.com/eoo/preparation. You can also find resources for Preparing Your Article for general guidance about writing and preparing your manuscript at www.wileyauthors.com/eoo/prepresources.

REVISION CHECKLIST:

We look forward to receiving your revised submission.

Yours sincerely,

Paul Greenhaff
Senior Editor
The Journal of Physiology

REQUIRED ITEMS

- Include a Key Points list in the article itself, before the Abstract.
- Author photo and profile. First or joint first authors are asked to provide a short biography (no more than 100 words for one author or 150 words in total for joint first authors) and a portrait photograph. These should be uploaded and clearly labelled together in a Word document with the revised version of the manuscript. See Information for Authors for further details.
- The Journal of Physiology funds authors of provisionally accepted papers to use the premium BioRender site to create high resolution schematic figures. Follow this link and enter your details and the manuscript number to create and download figures. Upload these as the figure files for your revised submission. If you choose not to take up this offer, we require figures to be of similar quality and resolution. If you are opting out of this service to authors, state this in the Comments section on the Detailed Information page of the submission form. The link provided should only be used for the purposes of this submission. Authors will be charged for figures created on this premium BioRender account if they are not related to this manuscript submission.
- Please upload separate high-quality figure files via the submission form.
- Please ensure that the Article File you upload is a Word file.
- Please include an Abstract Figure file, as well as the Figure Legend text within the main article file. The Abstract Figure is a piece of artwork designed to give readers an immediate understanding of the research and should summarise the main conclusions. If possible, the image should be easily 'readable' from left to right or top to bottom. It should show the physiological relevance of the manuscript so readers can assess the importance and content of its findings. Abstract Figures should not merely recapitulate other figures in the manuscript. Please try to keep the diagram as simple as possible and

without superfluous information that may distract from the main conclusion(s). Abstract Figures must be provided by authors no later than the revised manuscript stage and should be uploaded as a separate file during online submission labelled as File Type 'Abstract Figure'. Please also ensure that you include the figure legend in the main article file. All Abstract Figures should be created using BioRender. Authors should use The Journal's premium BioRender account to export high-resolution images. Details on how to use and access the premium account are included as part of this email.

Reviewing Editor's comments:

Two experts in the field reviewed this paper and found considerable value in the work but noted some deficiencies that should be corrected. I am sure the authors have no problem addressing the comments by the reviewers, which will help increase the impact of this paper.

Senior Editor's comments:

Thank you for the manuscript submission to The Journal of Physiology, which has been considered by a reviewing editor and two expert reviewers. Overall the opinion of the manuscript was positive and all were of the opinion that the manuscript contains interesting and novel data. However, a number of concerns have been identified by both reviewers that collectively detract from the scientific impact of the manuscript that the authors need to address.

Regards formatting of the manuscript, please note      the quality of "Fig. 1 Study design" was not at all clear in the pdf file submitted. Secondly, in line with Journal of Physiology author guidelines, if the research study was registered (clause 35 of the Declaration of Helsinki) the registration database should be indicated in the Methods, else the lack of registration should be noted as an exception (e.g. The study conformed to the standards set by the Declaration of Helsinki, except for registration in a database.).   Finally, The Journal of Physiology requires that nucleic acid and protein sequences, microarray data and data obtained using high throughput sequencing techniques, which support the results in the manuscript, should be archived in an appropriate public database (in guidelines to authors web page) and must be accessible without restriction from the date of publication.

We look forward to receiving the revised manuscript and rebuttal in due course.

Referee #1:

Re: JP-RP-2024-288368 "The skeletal muscle response to high-intensity training assessed by single-nucleus RNA-sequencing is blunted in individuals with type 2 diabetes" by Maria Hansen, Julius Elliot Raagaard Grothen, Anders Karlsen, Jaime Moreno Martinez, Nikos Sidiropoulos, Jørn Wulff Helge, Thomas Åskov Pedersen, and Flemming Dela

Overview: The present work sought to test muscle responses to exercise training in type 2 diabetes by RNA seq. Exercise training consisted of single leg extension, such that biopsies were obtained from trained and untrained legs. A control group was also included to compare training/no training responses to T2D. In short, the T2D group displayed little response to exercise compared with the control group, particularly in type IIa fibers. The findings suggest that T2D have blunted transcriptional responses to exercise.

Abstract

NA

Introduction

The authors might consider mentioning of work highlighting that T2D patients may have blunted clinical outcomes, such as insulin resistance/glucose tolerance, following exercise. This would align with some suggesting exercise resistance. Later in the discussion then, it would connect with ideas that longer training programs and/or alternative methods (e.g. nutrition/medication) may be needed for more aggressive approaches to manage blood glucose.

Method and Results

It is much appreciated that authors acknowledge using muscle tissue from a prior published work. However, it would be appropriate to describe the exercise intervention more opposed to having readers locate another paper. To this end, details on the clamp would be appreciated. For instance, what metabolic control prior to the clamp was implemented (e.g. control diet, avoid medications, etc.).

For the isoglycemic clamp, how much time elapsed from initial glucose reading to actual start of the clamp was started? Asking to think, the longer duration of time may lower blood glucose, and the isoglycemic clamp may overestimate insulin sensitivity. Then, was the time differences comparable between T2D and controls.

A major issue with this study is no females were included and a rationale is not provided. This limits understanding and generalization of findings.

Was the exercise prescription comparable between T2D and controls? Would imagine so since matched on fitness/BMI, but clear statements of this would be helpful.

How as diet controlled throughout the study?

Do authors anticipate medication selection of T2D may have influenced responses?

Was baseline differences tested? If not, this should be considered since it could influence responses to exercise between the groups.

It is mentioned that insulin-stimulated glucose clearance was ~30% higher compared with the untrained leg in both groups. However, as a new reader without prior knowledge of 2019 paper, it is not clear which stage this refers to. Also, the methods do not indicate how glucose clearance is calculated (e.g., is this GIR divided by ambient glucose levels or was insulin concentrations during the clamp considered)?

A minor point, but at times T2D is abbreviated and others it is spelled out. The same is true for some others (e.g. IF vs. immunofluorescence). Overall, would simply pick one and be consistent. Given the number of short-hands used, spelling out would be appreciated most likely by readers for perhaps less familiar terms.

Correlations were mentioned but not clear what was assessed.

Figure 1 is blurry and a little challenging to see.

In general, the figures at times are difficult to discern if there are differences or not. Is it possible at times for symbols to be used to denote differences?

Fig 4 has a correlation yet no p-value. Please include. Also, did any outcome measured herein, correlate with insulin sensitivity, blood flow, RQ/mitochondria work from the 2019 paper? If not, this goes to the discussions and meaning behind these data and prior functional/clinical work.

Discussion

The findings overall are very interesting, particularly the lack of transcriptional responses. It is of interest though that the 2019 paper shows no differences in insulin-stimulated glucose clearance. This raises the question - does the lack of transcriptional response matter for insulin sensitivity?

With comments on fiber switching, it is curious to hear if the authors view the data as supporting actual fiber switching, or if the findings suggest type II fibers have more characteristic of type I fibers. Fundamentally these are different processes despite generally it be discussed interchangeably. Seems important since Leg RQ from prior 2019 shows no differences, yet transcription may indicate less response in T2D. This ought to be discussed in context of fuel use and current findings seemingly being disconnected.

The term "cardiomyopathy" is used regarding discussion of type IIx fibers. Do the author contend that this relates to these fibers potentially having increased angiogenesis for blood flow? The clarity of this point is a bit unclear since the term is often used when discussing cardiac function. It would be helpful to connect more here since the prior study also shows blood flow results (healthy = higher blood flow than T2D). So what the comparative findings of the present work mean in the context of different blood flow would be helpful to discuss.

The lack of responses in T2D (page 13) related to glycogenolysis/glycolysis are interesting. In the 2019 paper, glycogen and RQ were not different. Subsequently, what do these results functionally mean? It seems with other measures no differences were observed. However, it is recognized that maximal workload was different from UT leg, albeit lower than controls.

Referee #2:

The objective of this study is to utilize single-nucleus RNAseq to understand myofiber adaptation to high intensity interval training (HIIT) in the context of type 2 diabetes. The primary advance of this study is the use of snRNAseq which captures the myonuclear transcriptional response which is obscured in bulk RNAseq and absent in single cell RNAseq. This tool is highly appropriate for this study as it enables transcriptional profiling of different fiber types, which are well recognized to respond differently to HIIT, and this work is the first to apply it to diabetic muscle. The high numbers of samples per group and the demonstration that fiber type distribution estimates from snRNAseq correlate with histology are additional strengths of the study.

My primary critique/concern is that the contention that "skeletal muscle response to high-intensity training... is blunted in individuals with type 2 diabetes" is not well supported. First because no other outcomes from the larger study utilizing these participants support this conclusion. Direct measures of the muscle's metabolic response to HIIT - particularly increased glucose clearance, glycogen breakdown and resynthesis - were equally evident in the diabetic subjects compared with control. As the differential gene expression between the control and diabetic participants centers on these pathways, this suggests that these baseline transcriptional differences don't translate to metabolic deficiencies impacting a training bout. Second, the transcriptional responses to HIIT training in the control group are relatively mild. Only a couple of genes in the metabolic and insulin related pathways have a log₂ fold change greater than 0.5. While I fully recognize that such minor

changes can have a substantial physiological effect, they will also be harder to detect in a variable dataset. It is not clear in this analysis whether none of the T2D samples exhibited transcriptional metabolic responses similar to the CON or whether the responses were too variable to reach statistical significance. Addressing this will be important to supporting the contention that the response is actually blunted in individuals with diabetes. Suggestions to address these and other critiques are outlined below:

1. A deeper discussion for the context of these findings with the previously published measures in these individuals is needed, and, by extension, more discussion related to how baseline differences in expression of metabolic genes should be interpreted in the context of metabolic health and exercise response.
2. It should be highlighted earlier that the study examines the chronic effects of HIIT, rather than the response to a bout following a training paradigm.
3. Please add some additional analyses of the genes of interest across the T2D and CON groups in response to training. Is the response truly "completely absent"? If the T2D group exhibits a variable response, are there any demographic or other factors to explain the variability?
4. Please provide the average biopsy cross-section or total count for each fiber type analyzed for histology to assess whether the sampling volume was similar between snRNAseq and histology. It would also be interesting to know whether some of the "unexplained" variance in Fig 4C could be explained by differences in cross-sectional area between the fiber types (under the assumption that smaller fibers have fewer myonuclei). If so, the prediction capacity of snRNAseq for fiber type distribution could break down in conditions with fiber type specific atrophy.
5. Figure 4D should be described in the Results. Is the change from UT to T in CON significant?
6. Overall, the graphical depictions of data are very clear and informative, but the outline colors for the different pathways in Figure 6 is hard to see with the fill colors. I think readability will be enhanced by removing or lightening the fill on the labels. Also a different color for "gluconeogenesis" would be helpful since that shade of blue is used to denote a group.
7. Having matched snRNAseq and histology info from so many biopsies is quite valuable. Consider extending the question of validating fiber types to other cell types by immunostaining for fibroblasts, macrophages and satellite cells in this or a subsequent study.

END OF COMMENTS

Comments from the Senior Editor and Referees

Senior Editor's comments:

Thank you for the manuscript submission to The Journal of Physiology, which has been considered by a reviewing editor and two expert reviewers. Overall the opinion of the manuscript was positive and all were of the opinion that the manuscript contains interesting and novel data. However, a number of concerns have been identified by both reviewers that collectively detract from the scientific impact of the manuscript that the authors need to address.

We thank the Senior Editor and both the reviewers for the time and effort in evaluating our manuscript. We appreciate the comments and suggestions and have adapted the manuscript accordingly. Below you'll find specific replies to each of the issues raised.

The manuscript has been thoroughly revised, including incorporating the information originally provided as supplementary information into the manuscript.

Regards formatting of the manuscript, please note the quality of "Fig. 1 Study design" was not at all clear in the pdf file submitted.

Figure 1 has been improved.

Secondly, in line with Journal of Physiology author guidelines, if the research study was registered (clause 35 of the Declaration of Helsinki) the registration database should be indicated in the Methods, else the lack of registration should be noted as an exception (e.g. The study conformed to the standards set by the Declaration of Helsinki, except for registration in a database.).

The exception has been added to the Methods p. 5 (version with tracked changes).

Finally, The Journal of Physiology requires that nucleic acid and protein sequences, microarray data and data obtained using high throughput sequencing techniques, which support the results in the manuscript, should be archived in an appropriate public database (in guidelines to authors web page) and must be accessible without restriction from the date of publication.

The data is being submitted to the EMBL Array Express database using the Annotare tool. <https://www.ebi.ac.uk/fg/annotare/> . We are waiting for a link and will forward it as soon as possible. This should of course be added to the manuscript..

We look forward to receiving the revised manuscript and rebuttal in due course.

Referee #1:

Re: JP-RP-2024-288368 "The skeletal muscle response to high-intensity training assessed by single-nucleus RNA-sequencing is blunted in individuals with type 2 diabetes" by Maria Hansen, Julius Elliot Raagaard Grothen, Anders Karlsen, Jaime Moreno Martinez, Nikos Sidiropoulos, Jrn Wulff Helge, Thomas ukov Pedersen, and Flemming Dela

Overview: The present work sought to test muscle responses to exercise training in type 2 diabetes by RNA seq. Exercise training consisted of single leg extension, such that biopsies were obtained from trained and untrained legs. A control group was also included to compare training/no training responses to T2D. In short, the T2D group displayed little response to exercise compared with the control group, particularly in type IIa fibers. The findings suggest that T2D have blunted transcriptional responses to exercise.

Abstract

NA

Introduction

The authors might consider mentioning of work highlighting that T2D patients may have blunted clinical outcomes, such as insulin resistance/glucose tolerance, following exercise. This would align with some suggesting exercise resistance. Later in the discussion then, it would connect with ideas that longer training programs and/or alternative methods (e.g. nutrition/medication) may be needed for more aggressive approaches to manage blood glucose.

We acknowledge that some studies have found blunted effects of training in type 2 diabetes. We have added this fact to the Introduction p. 4 (version with tracked changes). However, 100% of the participants in the present study had improvements in both respiratory capacity and glucose clearance.

Method and Results

It is much appreciated that authors acknowledge using muscle tissue from a prior published work. However, it would be appropriate to describe the exercise intervention more opposed to having readers locate another paper. To this end, details on the clamp would be appreciated. For instance, what metabolic control prior to the clamp was implemented (e.g. control diet, avoid medications, etc.).

We have elaborated upon the training program and clamp procedure on p. 5 (version with tracked changes).

For the isoglycemic clamp, how much time elapsed from initial glucose reading to actual start of the clamp was started? Asking to think, the longer duration of time may lower blood glucose, and the isoglycemic clamp may overestimate insulin sensitivity. Then, was the time differences comparable between T2D and controls.

For both groups we obtained arterial blood samples at 30 and 15 min before the clamp was initiated, and the level of glycaemia was determined as the average of the two.

A major issue with this study is no females were included and a rationale is not provided. This limits understanding and generalization of findings.

This is correct. In the Strengths and limitations sections we wrote: “*Limitations to the study design have been covered in the original publication (Dela et al., 2019)*”. In that publication, the limitation of including only males in the study is mentioned.

Was the exercise prescription comparable between T2D and controls? Would imagine so since matched on fitness/BMI, but clear statements of this would be helpful.

Yes, both groups exercised at the same relative workload, starting at 70% of the maximal one-legged workload and increasing until reaching a heart rate >80% of the maximal heart rate during the last two intervals. This has been added to the Methods p. 4 (version with tracked changes). However, as noted in the Discussion, the absolute workload was lower in the diabetes group.

How was diet controlled throughout the study?

One advantage of the one-legged training regimen is that the diet is not a confounder. However, between the groups, the diet may have differed and the diet was not controlled, except that the participants were instructed to eat a diet containing minimum 250 grams of carbohydrates/day in the 3 days leading up to the clamp experiment. This has been added to the Methods p. 5 (version with tracked changes).

Do authors anticipate medication selection of T2D may have influenced responses?

None of the participants changed medication during the training period, and the medication had been unchanged for at least 2 months before the inclusion in the study.

Two of the healthy control subjects were treated with antihypertensive medication (calcium ion antagonist, Amlodipine 5 and 10 mg daily, respectively). The patients with T2D were receiving this medication:

	Glucose lowering treatment (in addition to diet)	Antihypertensive drugs	Cholesterol-lowering drugs	Other
# 1		- Felodipine (Ca ⁺⁺ ion antagonist), 5 mg x 1 - Losartan/HCTZ (combined Angiotensin II receptor blocker and diuretic), 50 + 12,5 mg x 1	- Statin, Simvastatin 40 mg x 1	
#2	- Metformin (biguanide) 500 mg x1 - Amaryl (Glimepiride) 4 mg x 1			- Aspirin (acetylsalicylic acid) 75 mg x 1
#3	- Metformin (biguanide) 1000 mg x 2 - Victoza (GLP-1 agonist), 1,2 mg s.c. x 1			
#4	- Metformin (biguanide) 1500 mg x 1		- Statin, Simvastatin 40 mg x 1	
#5	- Metformin (biguanide) 1000 mg x 2	- Enalapril/HCTZ (combined Angiotensin II receptor blocker and diuretic), 20 + 12,5 mg x 1	-	- Alfuzosin (BPH treatment) 10 mg x 1
#6	-	-	-	-
#7	- Metformin (biguanide) 500 mg x 2	- Enalapril/HCTZ (combined ACE inhibitor and diuretic), 20 + 12,5 mg x 1	- Statin, Atorvastatin 20 mg x 1	- Alfuzosin (BPH treatment) 10 mg x 1
#8	- Metformin (biguanide) 1000 mg x 2	Enalapril (ACE inhibitor), 5 mg x 1		
#9	- Janumet (Sitagliptin/Metformin) 50 + 850 mg x 1	Losartan (Angiotensin II receptor blocker), 50 mg x 1 Centryl – Potassium (diuretic), 2,5 + 573 mg	Statin, Simvastatin 40 mg x 1	Aspirin (acetylsalicylic acid) 75 mg x 1

		x 1		1
#10	- Metformin (biguanide) 1000 mg x 2			

Altogether 9 patients with T2D received glucose-lowering agents, 5 received antihypertensive agents, and 4 received statins. This is a very common treatment regimen for these patients. However, it cannot a priori be excluded that the pharmacological treatment may have influenced the responses, but the design of the study (one-legged training) makes this unlikely.

Was baseline differences tested? If not, this should be considered since it could influence responses to exercise between the groups.

We did not perform clamp studies or obtain muscle biopsies before the training intervention started. The design of the study was such that one leg did not perform any training, while the other was trained. Thus, the non-training leg is considered as "baseline". These data are reported in the Results section and shown in Fig. 6 A and D, Fig. 7 A and C. The data are discussed in the paragraph entitled "*Transcriptional differences between T2D and CON in the untrained leg*" p. 13-14 (version with tracked changes).

There is a slight possibility/risk for a "carry-over" effect from the trained leg to the untrained leg. This would be very difficult to detect because it would require that we had obtained muscle biopsies from both legs before the one-legged training started. If we had done that there would be a much higher risk of introducing transcriptional changes in response to the biopsy procedure.

It is mentioned that insulin-stimulated glucose clearance was ~30% higher compared with the untrained leg in both groups. However, as a new reader without prior knowledge of 2019 paper, it is not clear which stage this refers to.

We have now included the specific data of the two stages in CON and T2D (Results, p. 9 (version with tracked changes))

CON:

- Stage 1 30.8%
- Stage 2 28.7%

T2D:

- Stage 1 32.6%
- Stage 2 24.0%

Also, the methods do not indicate how glucose clearance is calculated (e.g., is this GIR divided by ambient glucose levels or was insulin concentrations during the clamp considered)?

Leg glucose clearance was calculated as plasma flow (blood flow × (1-arterial haematocrit)) × ((Arterial-Venous)/Arterial) glucose concentrations divided by kg leg muscle mass.

The prevailing insulin concentration was similar between the CON and T2D was not included in the calculations:

Insulin (pM) [mean ± SEM]	CON	T2D
-----	-----

Baseline	33 ± 5	45 ± 8
Stage 1	872 ± 67	921 ± 90
Stage 2	11325 ± 545	10103 ± 675

(from Dela et al, 2019, Tables 1 and 4).

The method for calculating leg glucose clearance is now written on p. 5 (version with tracked changes).

A minor point, but at times T2D is abbreviated and others it is spelled out. The same is true for some others (e.g. IF vs. immunofluorescence). Overall, would simply pick one and be consistent. Given the number of short-hands used, spelling out would be appreciated most likely by readers for perhaps less familiar terms.

We have now spelt out type 2 diabetes, control (group), and immunofluorescence instead of using abbreviations.

Correlations were mentioned but not clear what was assessed.

It is not quite clear to us what the reviewer refers to. The correlations are described in the “*Statistical analyses*” section (*The squared Pearson’s coefficient and corresponding p-value were calculated for correlations between the proportion of type I fibres in the IF analysis and type I myonuclei (out of the total sum of type I, IIA and IIX myonuclei) in the snRNA-seq dataset.*) on p. 8-9 and in the “*Results*” section (*The proportion of type I fibres in the IF analysis correlated with the proportion of type I myonuclei identified in the snRNA-seq dataset ($R^2=0.30$, $p=0.0005$, Fig. 5C)*) on p. 10 (version with tracked changes).

Figure 1 is blurry and a little challenging to see.

The figure has now been improved.

In general, the figures at times are difficult to discern if there are differences or not. Is it possible at times for symbols to be used to denote differences?

The red arrow in Fig. 5D indicates a significant difference as noted in the legend. In Figs. 6 and 7, we show only differentially expressed genes (DEGs) which by definition are significant. This definition has now been made clearer in the Methods (p.8, version with tracked changes).

Fig 4 has a correlation yet no p-value. Please include.

The p-value ($p=0.0005$) has now been included in Fig. 5 C.

Also, did any outcome measured herein, correlate with insulin sensitivity, blood flow, RQ/mitochondria work from the 2019 paper? If not, this goes to the discussions and meaning behind these data and prior functional/clinical work.

We acknowledge that investigating possible associations between transcriptomic changes and physiological outcomes could provide insights into the mechanisms underlying skeletal muscle insulin sensitivity. However, we have some reservations about the robustness and interpretability of such an analysis in this specific context for the following reasons:

Limited Number of DEGs: Our single-nucleus RNA-sequencing data revealed a relatively small number of DEGs between trained and untrained muscles as well as between individuals with type 2 diabetes (T2D) and healthy controls. Given this limited transcriptional

response, correlating individual gene expression levels with insulin sensitivity measures would lack statistical power and could lead to spurious associations.

Multiple Comparisons & Statistical Concerns: Correlating a large number of individual DEGs with skeletal muscle glucose clearance would introduce a multiple comparisons problem, substantially increasing the likelihood of false positive findings. Even with appropriate statistical corrections (e.g., Benjamini-Hochberg FDR adjustment), the small number of DEGs may not yield meaningful or biologically relevant correlations.

Mismatch Between Molecular and Functional Measurements: The physiological measure of skeletal muscle insulin sensitivity (glucose clearance) is a complex, integrative outcome influenced by multiple factors beyond gene expression, including post-translational modifications, fibre-type composition, muscle capillary perfusion, and systemic insulin action. Transcriptomic changes at the single-nucleus level do not necessarily translate into functional metabolic outcomes at the whole-tissue or systemic level.

Temporal and Contextual Disparities: The RNA-seq data provide a snapshot of transcriptional activity, while insulin sensitivity reflects a dynamic functional process influenced by multiple regulatory mechanisms. The observed insulin sensitivity changes after training may be more dependent on epigenetic regulation, protein activity, or metabolic flux rather than acute changes in gene expression.

Given these limitations, we believe that a gene-by-gene correlation analysis would not be methodologically sound or biologically meaningful in this context. Instead, a more informative approach would be to assess whether broader transcriptional programs or pathways (e.g., insulin signaling, mitochondrial function) show trends that align with the physiological findings. Our analysis already includes pathway enrichment approaches to provide insight into potential mechanistic links.

Discussion

The findings overall are very interesting, particularly the lack of transcriptional responses. It is of interest though that the 2019 paper shows no differences in insulin-stimulated glucose clearance. This raises the question - does the lack of transcriptional response matter for insulin sensitivity?

We are slightly puzzled by your sentence “*It is of interest though that the 2019 paper shows no differences in insulin-stimulated glucose clearance*”, because there were marked differences between the two groups in whole-body glucose clearance and marked effects of training.

Nevertheless, in response to your specific question: We have tried to discuss this issue in the discussion (*HIIT-induced metabolic transcription occurs primarily in type IIA myonuclei and is blunted in T2D*, p. 16 (version with tracked changes)), and the answer is probably that insulin sensitivity is a complex, integrative outcome influenced by multiple factors beyond gene expression, including post-translational modifications, fibre-type composition, muscle capillary perfusion, and systemic insulin action. Epigenetic regulation, protein activity, or metabolic flux may also play a much more prominent role.

With comments on fiber switching, it is curious to hear if the authors view the data as supporting actual fiber switching, or if the findings suggest type II fibers have more characteristic of type I fibers. Fundamentally these are different processes despite generally

it be discussed interchangeably. Seems important since Leg RQ from prior 2019 shows no differences, yet transcription may indicate less response in T2D. This ought to be discussed in context of fuel use and current findings seemingly being disconnected.

The data indicate a transcriptional change from type IIX to IIA in the trained legs of the control group, which could – possibly – indicate a fibre type shift from type IIX to IIA. We see no transcriptional changes between type I and II. While your point on Leg RQ and the different preferences for fuel use in different fibre types is valid, this probably has no effect on the resting muscle. Additionally, we do not know of a method that could accurately measure this.

The term "cardiomyopathy" is used regarding discussion of type IIX fibers. Do the author contend that this relates to these fibers potentially having increased angiogenesis for blood flow? The clarity of this point is a bit unclear since the term is often used when discussing cardiac function. It would be helpful to connect more here since the prior study also shows blood flow results (healthy = higher blood flow than T2D). So what the comparative findings of the present work mean in the context of different blood flow would be helpful to discuss.

The KEGG pathway term "cardiomyopathy" appears in our analysis due to the differential expression of genes encoding structural and contractile proteins in skeletal muscle, including ACTN2, ACTN3, TPM1, TPM3, TTN, MYH7, and ATP2A1/ATP2A2. These genes are well-described in the context of skeletal muscle function and remodelling.

We acknowledge that the term "cardiomyopathy" is primarily associated with cardiac disease. However, in the context of skeletal muscle transcriptomics, this KEGG pathway term likely reflects structural adaptations in response to training rather than any direct implication for cardiac function. This highlights a known limitation of KEGG pathway annotations, as pathways identified in cardiac disease studies may also capture relevant gene expression changes in skeletal muscle.

Regarding the reviewer's mention of angiogenesis and blood flow, our interpretation of the KEGG "cardiomyopathy" pathway does not suggest changes in angiogenesis or perfusion. Our previous study showed that leg blood flow was similar between healthy controls and individuals with T2D and was consistently higher in the trained vs. untrained leg during insulin infusion ($P < 0.02$). We addressed the limitations of KEGG pathways in the "Strengths and Limitations" section: "*Lastly, the KEGG pathways, while informative, have built-in limitations such as only annotating pathways that have been previously curated. This was probably what led to the apparent effect of HIIT on cardiomyopathy-related KEGG pathways in our dataset*". Thus, the cardiomyopathy pathway findings should be interpreted primarily in the context of muscle remodelling rather than differences in blood flow or vascular adaptations. We have clarified this point in the revised manuscript in these two sections of the discussion: "Immunofluorescence fibre type distribution confirms the distribution of snRNA-seq myonuclei" on p. 13, where we first comment on the results, and "Strengths and limitations" on p. 17 (version with tracked changes).

The lack of responses in T2D (page 13) related to glycogenolysis/glycolysis are interesting. In the 2019 paper, glycogen and RQ were not different. Subsequently, what do these results functionally mean? It seems with other measures no differences were observed. However, it is recognized that maximal workload was different from UT leg, albeit lower than controls.

It is indeed an interesting question, which alludes to the overall, general minor transcriptional response in T2D. On p. 14-15, we have tried to discuss this finding in the context of other studies. There is a discrepancy between transcription and translation and on p. 16 (version

with tracked changes) we have now discussed this in three new paragraphs. As the reviewer mentions, glycogen content was similar between the two groups, and, in fact, so was glycogen synthase activity.

Referee #2:

The objective of this study is to utilize single-nucleus RNAseq to understand myofiber adaptation to high intensity interval training (HIIT) in the context of type 2 diabetes. The primary advance of this study is the use of snRNAseq which captures the myonuclear transcriptional response which is obscured in bulk RNAseq and absent in single cell RNAseq. This tool is highly appropriate for this study as it enables transcriptional profiling of different fiber types, which are well recognized to respond differently to HIIT, and this work is the first to apply it to diabetic muscle. The high numbers of samples per group and the demonstration that fiber type distribution estimates from snRNAseq correlate with histology are additional strengths of the study.

My primary critique/concern is that the contention that "skeletal muscle response to high-intensity training... is blunted in individuals with type 2 diabetes" is not well supported. Respectfully, we do not agree that the title ("*The skeletal muscle response to high-intensity training assessed by single-nucleus RNA-sequencing is blunted in individuals with type 2 diabetes*") of the manuscript is not well supported. In this experiment, we observed a surprisingly modest number of DEGs in the untrained leg between Controls and T2D, and similarly we found that only in type IIA myonuclei in the Controls (and not in T2D) significant DEGs were seen.

First because no other outcomes from the larger study utilizing these participants support this conclusion. Direct measures of the muscle's metabolic response to HIIT - particularly increased glucose clearance, glycogen breakdown and resynthesis - were equally evident in the diabetic subjects compared with control. As the differential gene expression between the control and diabetic participants centers on these pathways, this suggests that these baseline transcriptional differences don't translate to metabolic deficiencies impacting a training bout. Second, the transcriptional responses to HIIT training in the control group are relatively mild. Only a couple of genes in the metabolic and insulin related pathways have a log₂ fold change greater than 0.5. While I fully recognize that such minor changes can have a substantial physiological effect, they will also be harder to detect in a variable dataset. It is not clear in this analysis whether none of the T2D samples exhibited transcriptional metabolic responses similar to the CON or whether the responses were too variable to reach statistical significance. Addressing this will be important to supporting the contention that the response is actually blunted in individuals with diabetes. Suggestions to address these and other critiques are outlined below:

1. A deeper discussion for the context of these findings with the previously published measures in these individuals is needed, and, by extension, more discussion related to how baseline differences in expression of metabolic genes should be interpreted in the context of metabolic health and exercise response.

Regarding the concern that our conclusions are not supported by previous metabolic outcomes from this cohort, we acknowledge that both glucose clearance and glycogen turnover were increased in both groups following HIIT. However, our study specifically examines the transcriptional adaptations at the myonuclear level, providing novel insights into the differential gene expression responses between control and T2D individuals.

The observed baseline differences in metabolic gene expression between groups suggest inherent transcriptional distinctions that may not necessarily manifest as functional deficits under the conditions studied. It is important to emphasize that gene expression does not always directly translate to metabolic function due to post-transcriptional regulation, protein turnover, and other compensatory mechanisms.

Additionally, while the transcriptional response to HIIT in the control group was relatively modest, key metabolic pathways associated with glycogen breakdown and glycolysis showed coordinated upregulation in type IIA myonuclei, which was absent in the T2D group. This finding suggests that while both groups achieved similar improvements in glucose clearance, the transcriptional regulation underlying these adaptations differs. Whether this reflects an alternative mechanism of adaptation in T2D or a blunted transcriptional response remains an open question.

To further clarify this point, we have expanded the discussion to better contextualize these findings within previously published physiological data from this cohort and highlight the complexities of linking transcriptional regulation to metabolic outcomes (p. 16, version with tracked changes).

2. It should be highlighted earlier that the study examines the chronic effects of HIIT, rather than the response to a bout following a training paradigm.

We have now added this information in the first paragraph of the discussion, p. 12 (version with tracked changes). This is in addition to the more elaborate discussion of the timing of the biopsy, which follows at the end of the discussion, p. 17 (version with tracked changes) (*"As the biopsies were obtained 40 hours after the last exercise bout, we expect to have captured the more chronic effects of HIIT rather than the acute effects."*)

3. Please add some additional analyses of the genes of interest across the T2D and CON groups in response to training. Is the response truly "completely absent"? If the T2D group exhibits a variable response, are there any demographic or other factors to explain the variability?

We appreciate the suggestion to examine whether variability in T2D participants masked a potential transcriptional response. Explaining the variability can be challenging, as focusing on each sample to assess the training's impact on gene expression may not effectively demonstrate the differences. It is difficult to determine what degree of change constitutes a significant difference and what is merely noise sample by sample.

Therefore, we have now compared the variability of the control samples versus the type 2 diabetes samples, gene by gene. In short, we performed the F-test for all the seven genes that were significantly different in the trained vs untrained legs of the control group to assess if the variability between the control samples is similar to that of the diabetes group.

Our results (see figures below) indicate that the variability does not differ between the two groups across all seven genes. For transparency, we have included the relative expression levels of the seven relevant genes according to all conditions (control/diabetes, trained/untrained, and fibre type) in a new figure (Fig. 8). We acknowledge that describing

the response in the diabetes group as “completely absent” is likely too strong and have amended it to “blunted”.

F test results comparing the variability in the groups:

4. Please provide the average biopsy cross-section or total count for each fiber type analyzed for histology to assess whether the sampling volume was similar between

snRNAseq and histology. It would also be interesting to know whether some of the "unexplained" variance in Fig 4C could be explained by differences in cross-sectional area between the fiber types (under the assumption that smaller fibers have fewer myonuclei). If so, the prediction capacity of snRNAseq for fiber type distribution could break down in conditions with fiber type specific atrophy.

The total average count analyzed for histology is included in the Results: 494±221 fibres per sample (range 133–1,040). However, we have now added more detailed information comparing fibre types in a new table (Table 2). Please note that the immunofluorescence analysis contains only a distinction between type I and type II, and not type IIX and type IIA (in contrast to the snRNA-seq). To better compare the two, we have used type II myonuclei for the table calculated as the sum of type IIA and IIX myonuclei from the snRNA-seq analysis.

5. Figure 4D should be described in the Results. Is the change from UT to T in CON significant?

Figure 5D (former Fig. 4D) is described in the results, but we acknowledge that the placement was probably a bit confusing. The description has now been moved to fit with the order of the figures and the title of the section has been changed a little to accommodate this addition, p. 10 (version with tracked changes).

In the Figure legend, we have written that "The red arrow denotes a significant difference between the trained and untrained legs in CON."

6. Overall, the graphical depictions of data are very clear and informative, but the outline colors for the different pathways in Figure 6 is hard to see with the fill colors. I think readability will be enhanced by removing or lightening the fill on the labels. Also a different color for "gluconeogenesis" would be helpful since that shade of blue is used to denote a group.

Thank you for your positive feedback on our figures. Fig. 7 (former Figure 6) has been changed according to your suggestions.

7. Having matched snRNAseq and histology info from so many biopsies is quite valuable. Consider extending the question of validating fiber types to other cell types by immunostaining for fibroblasts, macrophages and satellite cells in this or a subsequent study.

Thank you for the suggestion. Regarding the present manuscript, we have chosen to restrict ourselves in data reporting, focusing primarily on myonuclei. We agree that it could prove interesting to perform similar analyses on the other cell types in the future.

Dear Professor Dela,

Re: JP-RP-2025-288368R1 "The skeletal muscle response to high-intensity training assessed by single-nucleus RNA-sequencing is blunted in individuals with type 2 diabetes" by Maria Hansen, Julius Elliot Raagaard Grothen, Anders Karlsen, Jaime Moreno Martinez, Nikos Sidiropoulos, Jørn Wulff Helge, Thomas Åskov Pedersen, and Flemming Dela

We are pleased to tell you that your paper has been accepted for publication in The Journal of Physiology.

Yours sincerely,

Paul Greenhaff
Senior Editor
The Journal of Physiology

If you would like to receive our 'Research Roundup', a monthly newsletter highlighting the cutting-edge research published in The Physiological Society's family of journals (The Journal of Physiology, Experimental Physiology, Physiological Reports, The Journal of Nutritional Physiology and The Journal of Precision Medicine: Health and Disease), please click this link, fill in your name and email address and select 'Research Roundup':
<https://www.physoc.org/journals-and-media/membernews>

- **TRANSPARENT PEER REVIEW POLICY:** To improve the transparency of its peer review process, The Journal of Physiology publishes online as supporting information the peer review history of all articles accepted for publication. Readers will have access to decision letters, including Editors' comments and referee reports, for each version of the manuscript as well as any author responses to peer review comments. Referees can decide whether or not they wish to be named on the peer review history document.
- You can help your research get the attention it deserves! Check out Wiley's free Promotion Guide for best-practice recommendations for promoting your work at: www.wileyauthors.com/eoo/guide. You can learn more about Wiley Editing Services which offers professional video, design, and writing services to create shareable video abstracts, infographics, conference posters, lay summaries, and research news stories for your research at: www.wileyauthors.com/eoo/promotion.
- **IMPORTANT NOTICE ABOUT OPEN ACCESS:** To assist authors whose funding agencies mandate public access to published research findings sooner than 12 months after publication, The Journal of Physiology allows authors to pay an Open Access (OA) fee to have their papers made freely available immediately on publication.

EDITOR COMMENTS

Reviewing Editor:

No further comments.

Senior Editor:

Thank you for the revised manuscript and comprehensive rebuttal article, which have been considered by the same reviewing editor and referees that considered the original submission. All are in agreement that the manuscript is now acceptable for publication and will be quite influential when published. Congratulations on a very nice study and thank you for considering The Journal of Physiology to publish your research.

REFeree COMMENTS

Referee #1:

No further comments.

Referee #2:

Thank you for your detailed response to my critiques. The edits to the manuscript sufficiently discuss the discrepancy between functional measures and transcriptional changes and my concerns have been addressed.